# Maternal inheritance of functional centrioles in two parthenogenetic nematodes

Aurélien Perrier[1], Nadège Guiglielmoni[2], Delphine Naquin [3], Kevin Gorrichon [4,5], Claude Thermes [3], Sonia Lameiras[6], Alexander Dammermann [7,8], Philipp H. Schiffer [2], Maia Brunstein[9], Julie C. Canman[10] & Julien Dumont [1] ✉

Centrioles are the core constituent of centrosomes, microtubule-organizing centers involved in directing mitotic spindle assembly and chromosome segregation in animal cells. In sexually reproducing species, centrioles degenerate during oogenesis and female meiosis is usually acentrosomal. Centrioles are retained during male meiosis and, in most species, are reintroduced with the sperm during fertilization, restoring centriole numbers in embryos. In contrast, the presence, origin, and function of centrioles in parthenogenetic species is unknown. We found that centrioles are maternally inherited in two species of asexual parthenogenetic nematodes and identified two different strategies for maternal inheritance evolved in the two species. In *Rhabditophanes diutinus*, centrioles organize the poles of the meiotic spindle and are inherited by both the polar body and embryo. In *Disploscapter pachys*, the two pairs of centrioles remain close together and are inherited by the embryo only. Our results suggest that maternally-inherited centrioles organize the embryonic spindle poles and act as a symmetry-breaking cue to induce embryo polarization. Thus, in these parthenogenetic nematodes, centrioles are maternally-inherited and functionally replace their sperm-inherited counterparts in sexually reproducing species.

Chromosome alignment and segregation in mitosis are orchestrated by a highly sophisticated spindle-shaped structure that is built from microtubules. Mitotic spindle and astral microtubules are primarily assembled from the centrosomes, comprised of two centrioles surrounded by the pericentriolar material (PCM), within which microtubule nucleation and anchoring occurs (Fig. 1a). Centrioles can also act as a symmetry-breaking cue to establish embryonic polarity in asymmetrically dividing embryos by defining the posterior pole of the zygote or one-celled embryo[1]. Importantly, centriole inheritance and function during cell division have mostly been studied in sexually-reproducing animal models.

Two main modes of centriole inheritance have been described in sexually-reproducing species. First, and as initially observed by Boveri more than a century ago, female meiotic divisions are preceded by

[1]Université Paris Cité, CNRS, Institut Jacques Monod, F-75013 Paris, France. [2]Worm~lab, Institute for Zoology, University of Cologne, Cologne, NRW, Germany. [3]Université Paris-Saclay, CEA, CNRS, Institute for Integrative Biology of the Cell (I2BC), 91198 Gif-sur-Yvette, France. [4]Centre de Référence, d'Innovation, d'eXpertise et de transfert (CRefIX), US 039 CEA/INRIA/INSERM, Evry, France. [5]Centre National de Recherche en Génomique Humaine (CNRGH), Institut de Biologie François Jacob, Direction de la Recherche Fondamentale, CEA Evry, France. [6]Institut Curie, PSL University, ICGex Next-Generation Sequencing Platform, 75005 Paris, France. [7]Max Perutz Labs, Vienna Biocenter Campus (VBC), 1030 Vienna, Austria. [8]University of Vienna, Center for Molecular Biology, Department of Microbiology, Immunobiology and Genetics, 1030 Vienna, Austria. [9]Institut Pasteur, Université Paris Cité, INSERM, Institut de l'Audition, F-75012 Paris, France. [10]Columbia University Irving Medical Center; Department of Pathology and Cell Biology, New York, NY 10032, USA. ✉e-mail: Julien.dumont@ijm.fr

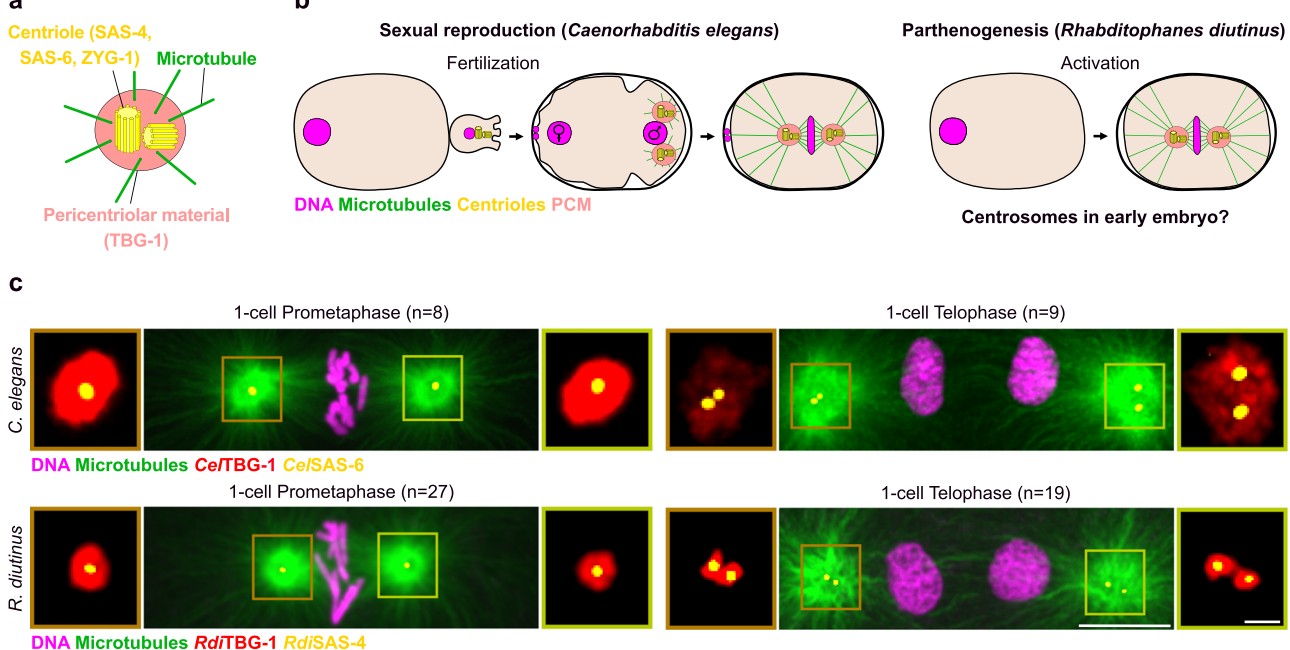

**Fig. 1 | Canonical centrosomes at mitotic spindle poles in the *R. diutinus* one-cell embryo. a** Schematic of a mature centrosome. Green, microtubules; light red, pericentriolar material; yellow, centrioles. **b** Schematics of the origin of one-cell embryo centrosomes in sexually-reproducing *Caenorhabditis elegans* (left) and in parthenogenetic *Rhabditophanes diutinus* (right). Magenta, DNA; green, microtubules; light red, pericentriolar material (PCM); yellow, centrioles. **c** Mitotic spindles during prometaphase (left) and telophase (right) in *C. elegans* (upper panels) and *R. diutinus* (lower panels) one-cell embryo. Magenta, DNA; green, microtubules; red, PCM; yellow, centrioles. Scale bar, 5 μm. Number of embryos examined is indicated at the top of each image for each stage. Insets, higher magnification of the centrosomes. Scale bar, 1 μm.

centrosome elimination occurring before diplotene of the prolonged prophase I[2–4]. Centrioles (and thus centrosomes) are usually paternally inherited at fertilization or formed de novo during early embryonic development as in rodents. Female meiotic spindles in most animal species, including in the Clade V nematode *Caenorhabditis elegans*[5], *Drosophila* and most vertebrates, are therefore self-organized in an acentrosomal and anastral manner that relies on chromosomes[6]. A second mode of centriole transmission has been described in several echinoderms, including sea urchins, starfish and sea cucumber, and in some mollusks such as surf clams and mussels[7–12]. In these marine species, the diakinesis oocyte contains a pair of centrosomes at the two spindle poles during meiosis I. A single centrosome is extruded in the first polar body, while the two centrioles of the second centrosome split to localize at the two poles of the meiosis II spindle. Following anaphase/telophase II, the centriole closest to the oocyte cortex is extruded in the second polar body, while a single daughter centriole remains in the zygote[13,14]. Despite the presence of this centriole of maternal origin, the zygotic spindle poles are exclusively formed by two centrosomes contributed paternally by the sperm during fertilization due to the rapid degeneration of the maternally-inherited centriole shortly after the completion of meiosis II[11,13–15]. Thus, in both modes of centriole inheritance, a transition from a meiotic to a mitotic mode of cell division occurs, either following fertilization when the sperm-supplied centrioles form the centrosomes[16–21], or later during early embryonic development by de novo centriole assembly[22].

In contrast, in animals that reproduce asexually by strict parthenogenesis, embryo development occurs without any male gamete contribution[23,24]. Despite this, in parthenogenetic hymenopteran insects, hexapods, and filarial nematodes, the zygotic spindle poles are decorated by large microtubule asters, indicative of centrosomal activity[25–29]. These large polar asters are sometimes preceded by cytoplasmic asters in oocytes, but the centrosomal origin of these asters has not been demonstrated; centrosomes have instead been proposed to

form de novo during early embryogenesis[26,28]. In any case, the origin of centrioles in such animals remain unknown. Furthermore, the maternal transmission of functional centrioles capable of assembling the zygotic spindle poles has not been documented in any species, regardless of their mode of reproduction, whether it be sexual or asexual through parthenogenesis. This has led to the widespread belief that paternal transmission or de novo assembly of functional centrioles are the only two plausible modes of centriole inheritance.

Here we have revisited this assumption by analyzing oocyte meiosis in two species of strict parthenogenetic nematodes belonging respectively to clade IV and V, namely *Rhabditophanes diutinus* and *Diploscapter pachys*. Through *R. diutinus* chromosome scale genome and transcriptome assembly, we identified orthologs of centrosomal and centriolar proteins against which we raised specific antibodies. We performed immunostaining experiments to show that zygotes of both species have canonical centrosomes, comprised of a pair of centrioles surrounded by pericentriolar material (PCM), at each spindle pole. By performing Ultrastructure Expansion coupled with Stimulated Emission Depletion (U-Ex-STED) microscopy[30,31] during oocyte meiosis in *R. diutinus*, we further discovered that, contrary to what is observed in *Drosophila* and vertebrates, but akin to echinoderms and mollusks, bona fide centrosomes, containing a pair of well-structured centrioles, persisted in oocytes throughout meiosis. However, unlike in echinoderms and mollusks, where the single maternally-transmitted centriole is degraded in the zygote, functional centrioles were maternally inherited in *R. diutinus* embryos. In both *R. diutinus* and *D. pachys* oocytes, a single polar body was extruded at the end of meiosis II, hence preserving zygotic diploidy in absence of fertilization. While a centrosome was extruded in this single polar body in *R. diutinus*, leading to the maternal transmission of a single centrosome to the zygote, both centrosomes were retained in the *D. pachys* zygote. Thus, different modes of maternal transmission of centrioles have evolved in different parthenogenetic nematode species. We also found that like in

sexual species, the position of the maternally-inherited centrioles correlated with the site of posterior zygotic polarization for asymmetric zygotic division in both species. Thus, our results suggest that the maternally-transmitted centrioles in these two asexual nematodes could potentially functionally replace their sperm-inherited counterparts in sexually reproducing species. Overall, our work represents the initial demonstration of maternal transmission of functional centrioles in any species, highlighting a mode of centriole inheritance that has previously been overlooked.

## Results

### *R. diutinus* embryos display canonical centrosomes

*R. diutinus* is a Clade IV free-living nematode closely related to the family of *Strongyloides* parasites[32]. *R. diutinus* are obligate parthenogenetic worms with exclusively asexual female individuals that completely lack male gametes[33]. To determine if the mitotic spindle, which assembles after parthenogenetic oocyte activation in *R. diutinus*, forms in a centrosomal or acentrosomal manner (Fig. 1b), we set out to localize centriolar and PCM components in *R. diutinus* embryos by immunolabeling. For this, we first re-sequenced and fully assembled a high-quality chromosome scale *R. diutinus* genome and transcriptome (Supplementary Fig. 1a–d). We identified and raised antibodies against orthologs of the centriolar structural protein CPAP/CENPJ (*R. diutinus* SAS-4) and the PCM component γ-tubulin (*R. diutinus* TBG-1) (Supplementary Fig. 2a, b and 4a, b, d)[34–36]. Immunofluorescence analysis of mitotic *R. diutinus* one-cell embryos showed that, similarly to in *C. elegans* (Fig. 1c, upper panel), prometaphase spindles (Fig. 1c, lower panel) displayed prominent astral microtubule structures at both poles. Each aster was centered on a puncta of centriolar proteins (marked by SAS-4 in *R. diutinus* and SAS-6 in *C. elegans*) surrounded by PCM (TBG-1)[37]. During mitosis in the *C. elegans* zygote, daughter and parent centrioles remain engaged until late in mitosis, after chromosome segregation[38]. The two centrioles of each pair then physically separated into two distinct puncta during telophase through a process termed centriole "disengagement"[20], which is required for the next centriolar duplication cycle[39]. Accordingly, in both *C. elegans* and *R. diutinus* zygotes during telophase, all foci of centriolar proteins were resolved into two distinct puncta corresponding to the two disengaged centrioles within each centrosome (Fig. 1c, right). Thus, despite the lack of fertilization by a male gamete, the spindle poles of *R. diutinus* embryos form from canonical centrosomes, each comprised of a pair of centrioles surrounded by PCM components.

### The gonad of *R. diutinus* differs from that of *C. elegans*

We envisioned two plausible scenarios to explain the presence of centrosomes in *R. diutinus* embryos: 1) rapid de novo formation of centrosomes after oocyte meiosis or 2) atypical maternal transmission of centrioles. To distinguish between these hypotheses, we first analyzed the germline organization of *R. diutinus* to identify the germline stem cells and oocytes. The gonad organization of *R. diutinus* parthenogens and *C. elegans* hermaphrodites appeared superficially similar with two symmetric folded arms, separated by two spermatheca structures, from a common central uterus containing developing embryos. However, while the two spermathecas contained mature sperm in *C. elegans* adult hermaphrodites, they were empty vestigial structures in *R. diutinus* that lacked DNA-containing foci. The *C. elegans* gonad arms were filled with nuclei of a relatively even size, which correspond to mitotically-dividing germline stem nuclei in the distal-most part of the gonad, followed by maturing oocytes progressing proximally toward the spermatheca (Supplementary Fig. 7a)[40]. In stark contrast, the gonad of *R. diutinus* was mostly filed with giant nuclei of varying size, except after the gonad turn in the region proximal to the vestigial spermatheca where evenly-sized small compact nuclei (SCN) were found (Fig. 2a and Supplementary Fig. 5a)[41]. This peculiar germline organization, with giant nuclei juxtaposed

to a zone filled with small compact nuclei[41], has previously been described for other Strongyloïd parasitic nematodes including *S. ratti, S. papillosus, S. stercoralis*, and *P. trichosuri*[33,42]. By combining filamentous actin staining and serial-block face scanning electron microscopy (SBF-SEM), we found that the smaller nuclei were densely packed in a common syncytial cytoplasm in this region, before being progressively cellularized into larger cells that resembled prophase I diplotene/diakinesis oocytes in *C. elegans* (Supplementary Fig. 5b, c). Immunofluorescence analysis using SAS-4 antibodies identified centrioles in this spermatheca-proximal region, with two foci associated with each nucleus (Fig. 2b). This pattern correlated with the occasional appearance of spindle-like structures (Fig. 2c), with SAS-4 at the poles (Supplementary Fig. 6a) and phospho-histone H3 (pH3) positive nuclei (Fig. 2d). Together, these results suggest that mitosis occurs in this small spermatheca-proximal portion region of the gonad, and that the pH3 positive small compact nuclei correspond to mitotically-dividing germline stem nuclei.

To further characterize cell cycle progression within the gonad structure in *R. diutinus*, we performed an EdU incorporation pulse-chase experiment (Supplementary Fig. 6b). We found that after a 1 h chase, the EdU signal was mostly constrained to the pH3-positive region (Supplementary Fig. 6c) and after a 28 h chase the EdU signal shifted toward the cellularized oocytes (Supplementary Fig. 6d)[43]. Further, immunofluorescence staining with an anti-pMPM-2 antibody (Fig. 2e), which recognizes various phosphorylated mitotic/meiotic epitopes[44], revealed five filament-like structures inside each nucleus in a zone directly adjacent to the mitotic region. This chromosomal axis-like pattern correlates with the *R. diutinus* karyotype, comprised of five chromosome pairs (Supplementary Fig. 1e), and is reminiscent of the synaptonemal complex, which assembles between homologous chromosomes in pachytene nuclei during early prophase I in *C. elegans* oocytes (Supplementary Fig. 6e)[45]. This observation suggests that, in *R. diutinus*, homologous chromosomes undergo synapsis in this mitotic region-adjacent zone. The small compact nuclei zone is thus subdivided into a distal mitotic zone, which contains germline stem nuclei, and a proximal meiotic transition zone, which is adjacent to the vestigial spermatheca (Fig. 2f). Meiotic nuclei are then cellularized into large oocytes, which transit through the vestigial spermatheca to become embryos without fertilization. These results suggest gonad structure differs dramatically between the asexually reproducing *R. diutinus* and the sexually reproducing *C. elegans*.

### Centrioles are maintained after diplotene in *R. diutinus*

In the *C. elegans* gonad, centrioles are present until the end of pachytene, but are eliminated abruptly at the diplotene stage (Supplementary Fig. 7b)[3]. In *R. diutinus*, SAS-4-positive foci were also visible near pachytene-like nuclei, but strikingly they persisted at later stages of diplotene and following cellularization during diakinesis (Supplementary Fig. 7d). SAS-4-positive foci were also detected in the last cellularized oocyte before the vestigial spermatheca (-1 oocyte). To confirm that the SAS-4 positive foci observed in oocytes were indeed centrioles, we raised antibodies against the *R. diutinus* ortholog of the centriolar protein PLK4 (*RdiZYG-1*) (Supplementary Fig. 3a and 4a, e)[46]. Immunofluorescence staining using these antibodies revealed colocalization with SAS-4 foci throughout the *R. diutinus* gonad, from the mitotic zone to the -1 diakinesis oocyte (Fig. 2g). Thus, unlike in *C. elegans* and mice, where centrioles are eliminated before entering the diplotene stage, but similar to in echinoderms and mollusks, centrioles are present throughout meiotic prophase I in parthenogenetic *R. diutinus*[2,3,47].

### Astral and centrosomal spindles form in *R. diutinus* oocytes

To determine if the centrioles present in *R. diutinus* diakinesis oocytes are maternally transmitted (Fig. 3a), we first analyzed the meiotic divisions by immunofluorescence. In *C. elegans*, as in most species,

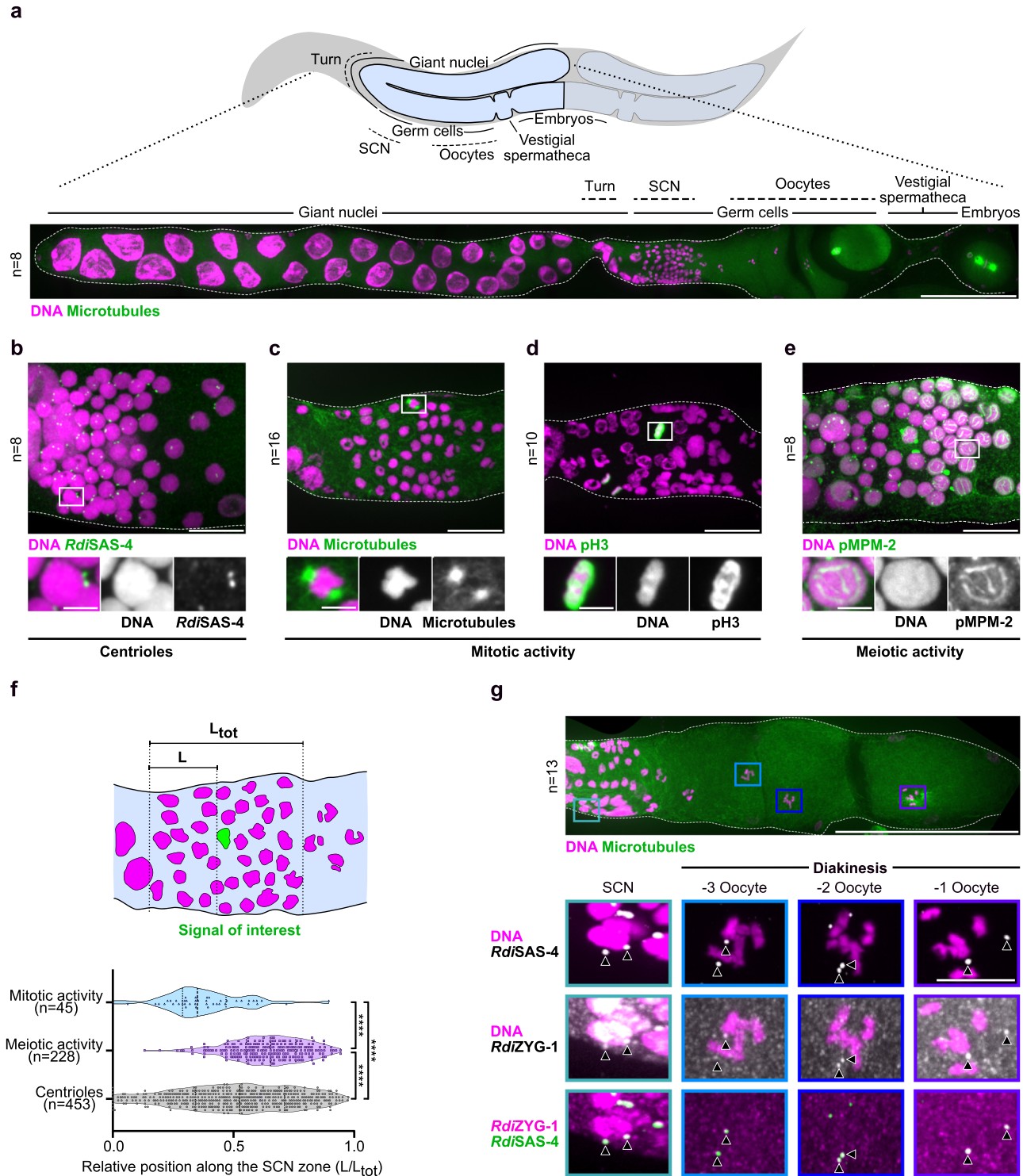

**Fig. 2 | Centrioles are present throughout the gonad in *R. diutinus*. a** Top, schematic of *R. diutinus* highlighting the reproductive system. Bottom, immuno-fluorescence image of a straightened *R. diutinus* gonad. Magenta, DNA; green, microtubules. The gonad is delimited by a white dashed line. Scale bar, 50 μm. Immunofluorescence images of *R. diutinus* gonads centered on the vestigial spermatheca-proximal region that contains the mitotically-dividing germline nuclei and early meiotic nuclei (leptotene to diplotene) and stained (green) for *Rdi*SAS-4 (**b**), microtubules (**c**), pH3 (**d**), pMPM-2 (**e**), and (magenta) DNA. Gonads in (**b**) and (**e**) were spread before staining. Scale bar, 10 μm. Bottom insets show higher magnification of one nucleus. Scale bar, 2 μm. Number of gonads examined is listed for each antibody used. **f** Schematic (top) of the spermatheca-proximal small

condensed nuclei (SCN) zone and plot (bottom) of the spatial distribution of mitotic activity (pH3), meiotic activity (pMPM-2), and centrioles (*Rdi*SAS-4) across the SCN zone from the distal to the spermathecal proximal region. Means and quartiles are shown. Kruskal-Wallis multiple comparison test, alpha = 0.05, ****$p \leq 0.0001$. **g** Top, immunofluorescence image of an *R. diutinus* gonad arm centered on the vestigial spermatheca-proximal region and the diakinetic oocytes. Scale bar, 50 μm. Bottom insets show higher magnification of one nucleus. Top row, magenta, DNA; greyscale, *Rdi*SAS-4. Middle row, magenta, DNA; greyscale, *Rdi*ZYG-1. Bottom row, magenta, *Rdi*ZYG-1; green, SAS-4. Black arrowheads indicate *Rdi*SAS-4 and/or *Rdi*ZYG-1 signal. Scale bar, 5 μm. Source data are provided as a Source Data file.

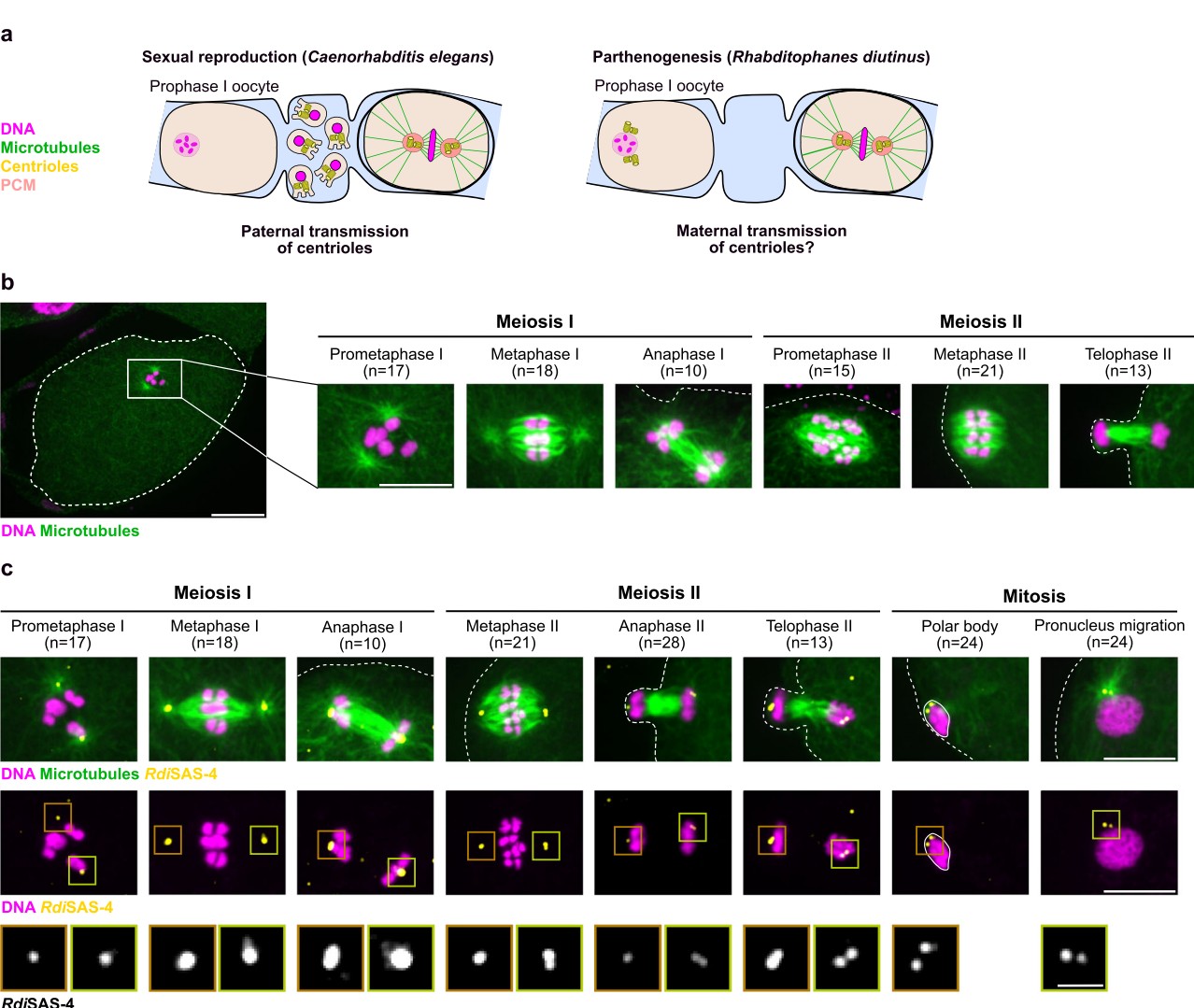

**Fig. 3 | Centriolar foci are present at the spindle poles throughout meiosis in *R. diutinus* oocytes. a** Schematics of the paternal and potential maternal origin of one-cell embryo centrosomes in sexually-reproducing *Caenorhabditis elegans* (left) and in parthenogenetic *Rhabditophanes diutinus* (right) respectively. Magenta, DNA; green, microtubules; yellow, centriole; light red, PCM. **b** Immunofluorescence image of an *R. diutinus* oocyte in prometaphase I. The oocyte contour is highlighted with a white dashed line. Scale bar, 10 µm. Number of oocytes examined is indicated at the top of the images for each stage. Images on the right show higher magnifications centered on the spindle during the indicated oocyte meiotic division steps. Magenta, DNA; green, microtubules. Scale bar, 5 µm.

**c** Immunofluorescence images centered on the spindle during the indicated oocyte meiotic division steps and on the polar body and maternal pronucleus during mitosis in *R. diutinus*. Magenta, DNA; green, microtubules; yellow, *Rdi*SAS-4. Scale bar, 5 µm. Oocyte and embryo contours are highlighted with a white dashed line. The polar body is highlighted by a white line. Number of oocytes or embryos examined is indicated at the top of the images for each stage. Bottom insets show higher magnifications of *Rdi*SAS-4 at the left (brown square) and right (green square) meiotic spindle poles or during polar body emission (brown square) and pronuclear migration (green square). Scale bar, 1 µm.

centrosomes are eliminated well before nuclear envelope breakdown in the oocyte. The meiosis I and II acentrosomal spindles, which form in absence of centrosomes, display a characteristic barrel-shape (Supplementary Fig. 8a, left), with no astral microtubules and flat spindle poles[16]. In contrast, *R. diutinus* prometaphase I/metaphase I oocyte spindles displayed prominent microtubule asters at both spindle poles (Supplementary Fig. 8a, right). These polar asters progressively disappeared as the oocyte progressed toward anaphase I, and did not reappear during meiosis II (Fig. 3b). Immunofluorescence analysis of the centriolar markers SAS-4 and ZYG-1 revealed that both proteins were present at the center of each polar aster (Fig. 3c and Supplementary Fig. 8d) and were surrounded by a significant accumulation of the PCM component TBG-1 (Supplementary Fig. 8b, c). Thus, mature centrosomes capable of nucleating astral microtubules are present at the meiotic spindle poles in prometaphase I/metaphase I oocytes in *R. diutinus*. Interestingly, the loss of astral microtubules observed as the

oocyte progressed toward anaphase I, and during meiosis II, was concomitant with a steep decrease in TBG-1 signal at the meiotic spindle poles. This could be due to a late centrosome elimination process after metaphase I or to a loss of nucleating capacity of the meiotic centrosomes in *R. diutinus* oocytes. Consistent with the second hypothesis, and in contrast to the total loss of TBG-1 and polar asters, low levels of SAS-4 and ZYG-1 remained at spindle poles throughout meiosis I and II (Fig. 3c and Supplementary Fig. 8d). After telophase II, two foci of the centriolar markers were clearly visible associated with the maternal pronucleus and in the polar body. Thus unlike in *C. elegans*, in *R. diutinus*, foci of centriolar proteins are present throughout oocyte meiosis and are maternally transmitted to the one-cell embryo.

### Abortive cytokinesis at the end of meiosis I in *R. diutinus*

Next, we analyzed meiotic polar body emission in *R. diutinus* and *C. elegans* oocytes. In *C. elegans* oocytes, the six meiosis I bivalent

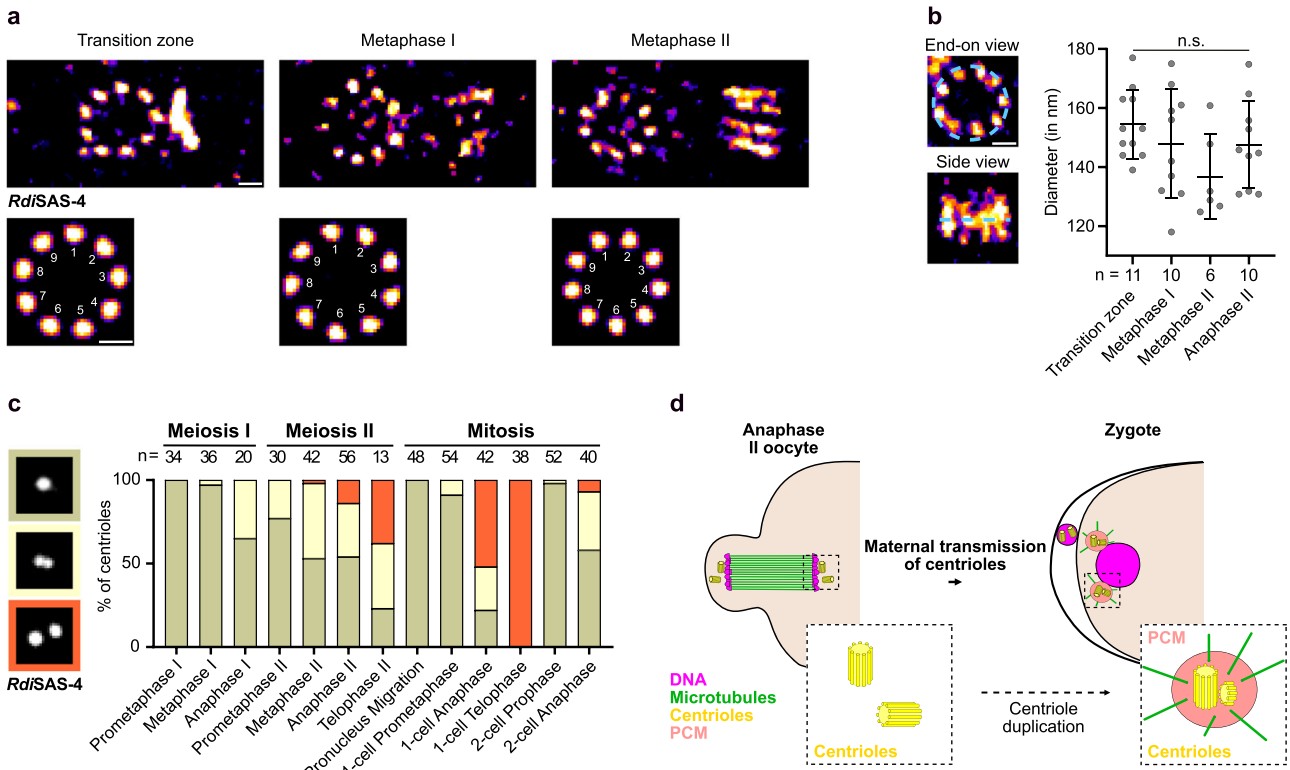

**Fig. 4 | Maternal inheritance of intact centrioles in the *R. diutinus* one-cell embryo. a** U-Ex-STED microscopy images of centrioles observed at *R. diutinus* oocyte spindle poles at different meiotic stages. *Rdi*SAS-4, fire lookup table. Corresponding bottom images were obtained after 9-fold symmetrization. Scale bar 0.2 μm. **b** Quantification of individual centriole diameters at different *R. diutinus* oocyte meiotic stages. Number of centrioles examined is indicated beneath each plot. An example of centriole "end-on view" and "side view" are displayed. *Rdi*SAS-4, fire lookup table. Scale bar 0.2 μm. Kruskal-Wallis multiple comparison test, alpha = 0.05, n.s. = *p* > 0.05. Mean and standard deviation are displayed.

**c** Quantification of the shape of the *Rdi*SAS-4 foci at the indicated meiotic and mitotic stages. Grey, light beige, and orange correspond to 1-spot, 2-spot adjacent/engaged, and 2-spot separated/disengaged foci respectively. Number of centrioles examined is indicated at the top of each bar. **d** Schematic of the maternal transmission of a single centrosome from the oocyte to the one-cell embryo in *R. diutinus*. Insets are schematics of the transmitted centriolar and centrosomal structures. Magenta, DNA; green, microtubules; yellow, centrioles; light red, PCM. Source data are provided as a Source Data file.

chromosomes and six meiosis II pairs of sister chromatids align at the metaphase plate, and are segregated in anaphase I and II respectively. This leads to the extrusion of six homologous chromosomes and six sister chromatids into two polar bodies at the end of meiosis I and II respectively, and to the formation of a single haploid mature oocyte essential for the success of sexual reproduction[48]. In contrast, we consistently observed a single polar body attached to early *R. diutinus* embryos. Meiosis I was invariably incomplete with a failure to extrude five homologous chromosomes into a first polar body (Supplementary Fig. 9a, b). As a consequence, ten pairs of sister chromatids were consistently observed realigning on the prometaphase II spindle. Aligned pairs of chromatids segregated into twenty sister chromatids during anaphase II, ten of which were extruded into the single meiosis II polar body. The failure or success of polar body extrusion correlated with the absence or presence of a filamentous actin (F-actin) ring at the base of the polar body at the end of meiosis I or II respectively (Supplementary Fig. 9c–e). Abortive cytokinesis also led to retention of centriolar SAS-4 foci, present at both meiotic spindle poles, inside the meiosis II oocyte. Thus, unlike in *C. elegans* oocytes, which extrude two polar bodies, failure to extrude the first polar body via abortive cytokinesis at the end of meiosis I in *R. diutinus* leads to the retention of two sets of chromosomes and two centriolar foci[49–51].

### A pair of centrioles is transmitted to *R. diutinus* embryos

To evaluate the nature of the centriolar foci maternally transmitted to the *R. diutinus* embryo, we performed U-Ex-STED microscopy on

*R. diutinus* gonads following SAS-4 immunostaining (Fig. 4a and Supplementary Fig. 8e). Consistent with recent findings in *C. elegans*, within the transition zone every SAS-4 focus near a meiotic nucleus appeared as a ring-shaped structure made of individual puncta organized in a characteristic nine-fold symmetrical configuration[31]. SAS-4 was also consistently found associated with the base of the forming procentriole (Fig. 4a and Supplementary Fig. 8e). The nine-fold symmetrical arrangement of the mother centriole remained evident throughout meiosis I, while daughter centriole length gradually increased, concurrent with its progressive disengagement and physical separation from its mother. By anaphase II, the mother and daughter centrioles were no longer oriented in an orthogonal configuration (Supplementary Fig. 8e). In the *C. elegans* germline, centriole widening marks the onset of centriole elimination[52]. In contrast, the diameter of the SAS-4 ring-shaped structure remained relatively constant throughout meiosis in *R diutinus* (Fig. 4b). Thus, intact centrioles are present throughout meiosis in *R. diutinus* oocytes and each SAS-4 focus in telophase II corresponded to an individual intact centriole. Furthermore, conventional light microscopy revealed that the proportion of engaged and separated double SAS-4 foci increased as meiosis progressed, such that by telophase II, the large majority of oocytes displayed two separated foci at each spindle pole (Fig. 4c). Combined, these results suggest that, unlike in echinoderms and mollusks where a single centriole is maternally transmitted to the zygote shortly before disintegrating[13,14], two disengaged intact centrioles are maternally transmitted to the one-cell *R. diutinus* embryo.

This situation mirrors the scenario observed in *C. elegans* zygotes, where two paternal centrioles are inherited. In *C. elegans*, following fertilization and preceding the first mitosis of the zygote, the zygotic genome undergoes a pre-mitotic S-phase, during which the two paternally-inherited centrioles are also duplicated. The two newly formed centrosomes, each comprised of a pair of centrioles, form the two poles of the zygotic spindle. The two centrioles of an engaged pair can be resolved as two foci following their disengagement during telophase (Fig. 1c)[20]. The observation of an identical centriolar pattern, with two disengaged centrioles visible at each disassembling spindle pole in one-cell telophase *R. diutinus* embryos (Fig. 1c), suggests that the two maternally-inherited *R. diutinus* centrioles have been duplicated during the pre-mitotic S-phase, just as sperm-derived centrioles do in *C. elegans*[37]. Furthermore, as in *C. elegans*, concomitant with duplication, the two newly formed centrosomes underwent a maturation phase as evidenced by the re-accumulation of the PCM component TBG-1 (Fig. 4d and Supplementary Fig. 8b, c). Altogether, these results provide the first demonstration of maternal transmission of functional centrioles during asexual reproduction in any species.

## Two pairs of centrioles are transmitted to *D. pachys* embryos

To assess the evolutionary conservation of the maternal inheritance of centrioles in parthenogenetic nematodes, we examined meiosis in another parthenogenetic nematode, *Diploscapter pachys*. *D. pachys* is a Clade V nematode with a single chromosome pair and which, as in *R. diutinus*, undergoes a single round of polar body extrusion[53,54]. The genomes of *D. pachys* and *C. elegans* are evolutionarily related enough that antibodies to *C. elegans* PCM and centriolar proteins can be successfully employed in *D. pachys*. Similar to what we observed in *R. diutinus*, two centriolar foci stained with ZYG-1 antibodies were associated with each nucleus along the entire gonad arm, from the mitotic zone at the distal tip of the gonad arm to diakinetic cellularized oocytes (Fig. 5a). However, in contrast to the meiotic spindles of *R. diutinus* where centrosomes formed each spindle pole, during metaphase I in *D. pachys*, the two centrosomes remained close to each other, between or adjacent to the chromosomes within the spindle, and did not organize the spindle poles or form astral microtubules (Fig. 5b). During anaphase II, the centrosomes were consistently found associated with the set of chromosomes that remained in the oocyte and became the maternal pronucleus. In line with this and in contrast to in *R. diutinus*, we never detected centrosomes in the single extruded polar body in *D. pachys* embryos. Furthermore, by immunostaining *D. pachys* oocytes using another centrosomal marker, Cep192 (SPD-2), we occasionally observed individual centrosomes resolved as adjacent double foci, suggesting that they each contained two centrioles (Fig. 5c)[55]. In contrast to in *R. diutinus*, we never observed centrosomes fully resolved as two separated disengaged foci. Together, these observations suggest that, as in *R. diutinus*, centrioles are maternally inherited in *D. pachys*. Yet, unlike in *R. diutinus* where a single pair of centrioles is maternally transmitted to the one-cell embryo, our results also suggest that *D. pachys* embryos inherit a pair of centrosomes, each comprised of two continuously engaged centrioles (Fig. 5d). Thus, different strategies have been employed in parthenogenetic nematodes to ensure maternal transmission of centrioles during reproduction.

## Inherited centrioles correlate with the zygote posterior pole

We next examined the implications of the maternal transmission of centrioles for embryo polarization. In *C. elegans*, the kinase Aurora A/AIR-1 concentrates at the sperm-inherited centrosome and induces antero-posterior polarization of the asymmetrically dividing zygote[56–59]. AIR-1 activity displaces actomyosin at the anterior cortex surrounding the maternal pronucleus, which induces ruffling of the anterior cortex and the spatial segregation of PAR polarity proteins (PAR-3/PAR-6/PKC-3 at the anterior and PAR-1/PAR-2 at the posterior)[60]. To determine if the maternally-inherited centrosome can act as the symmetry breaking cue to induce polarization of the asymmetrically dividing *R. diutinus* embryo (Fig. 6a), we imaged *R. diutinus* embryos throughout development by DIC microscopy. As in *C. elegans* embryos, the position of the centrosome in one-cell *R. diutinus* embryos always correlated with the development of the posterior blastomere and the tail in later stage embryos and larvae, respectively (Fig. 6b)[61]. This was confirmed by DIC analysis of the first two embryonic divisions at higher temporal resolution (Fig. 6c). In both *C. elegans* and *R. diutinus*, the first mitotic division led to the formation of two blastomeres of different size (Fig. 6d). However, their location relative to the initial position of the maternal pronucleus was inverted, instead correlating with the initial position of the centrosome. The larger anterior blastomere AB, which always divides before the smaller posterior P1 blastomere in *C. elegans* two-cell embryos, formed at the opposite end of the embryo in *R. diutinus* (Fig. 6e and Supplementary Fig. 10a)[62]. Thus, like in *C. elegans*, the position of the posterior blastomere in *R. diutinus* correlates with the initial location of the centrosome. Yet, because the maternally-inherited centrosome is at the opposite end of the embryo, the anterior-posterior polarity axis is inverted relative to the maternal pronucleus in *R. diutinus* compared to in *C. elegans*.

To determine the initial stages of polarization in *R. diutinus* one-cell embryos, we analyzed two early markers of anterior polarity: ruffling of the anterior one-cell embryo cortex (Fig. 6c) and the anterior cortical accumulation of F-actin (Fig. 6f, g). As expected, we could detect these two markers on the side opposite to the maternal pronucleus and its associated centrosome. These early signs of polarization were detectable only at the beginning of pronuclear migration of the maternal pronucleus but not at earlier stages, despite the presence of centrosomes throughout the meiotic divisions. This demonstrates that, as in *C. elegans*, the cortex above the unique centrosome in *R. diutinus* one-cell embryos acquires a posterior identity just prior to pronuclear migration toward the embryo center, but not before.

*C. elegans* oocytes can undergo spontaneous ectopic polarization during meiosis[56–59,63]. This premature polarization is normally inhibited by the Aurora A/AIR-1 kinase activity in the cytoplasm. After completion of the two meiotic divisions, the pair of sperm-derived centrioles acquires a thin shell of PCM, including a small amount of AIR-1. AIR-1 then turns into an activator of polarization of the zygote[57]. In *R. diutinus*, we found AIR-1 (Supplementary Fig. 3b and 4a, c) was present on centrioles throughout oocyte meiosis, yet signs of polarization only appeared at the pronuclear migration stage (Supplementary Fig. 10b, c). This late polarization could not be explained by the low level of centrosomal AIR-1 during meiosis, because AIR-1 was not further enriched at centrosomes during symmetry breaking, until the onset of centrosome maturation during prophase/prometaphase of mitosis.

Altogether, these results demonstrate that in parthenogenetic *R. diutinus*, the position of the maternally-inherited centrioles correlates with zygotic posterior polarization. This, combined with our results from spindle localization, suggests that maternally inherited centrioles have the potential to functionally replace their *C. elegans* sperm-inherited counterparts and act as a symmetry-breaking cue to induce polarization of the one-cell embryo after meiosis is completed.

## Discussion

Here, we show maternal inheritance of centrioles in parthenogenetic nematodes. This work represents a significant step toward a better understanding of the different modes of reproduction that have emerged during evolution. In all species studied to date, maternal centrioles are eliminated prior to the first zygotic division either before the oocyte meiotic divisions, as observed in vertebrates, *C. elegans*, and *Drosophila*, or during the meiotic divisions, as seen in certain echinoderms and mollusks[47]. Thus, our findings in two parthenogenetic

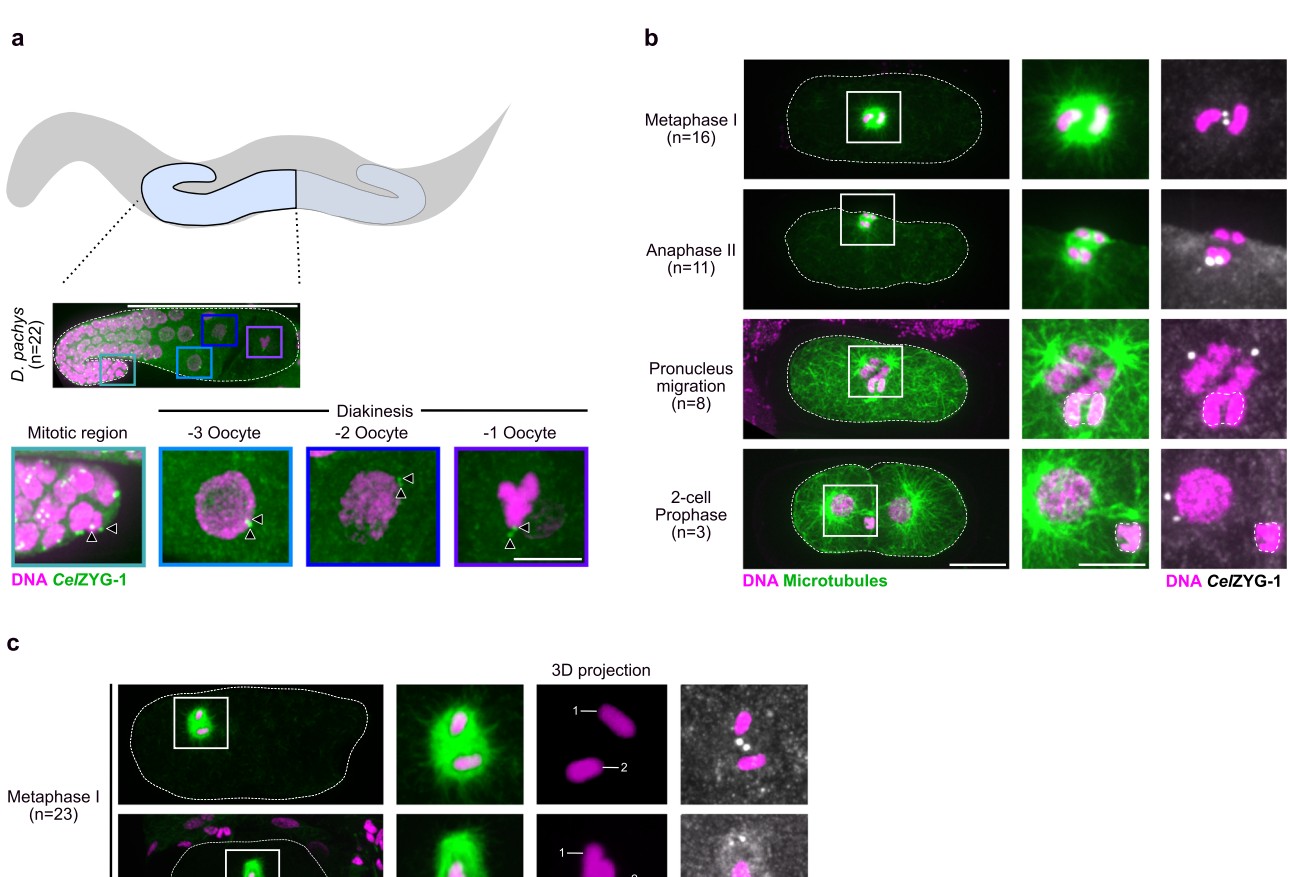

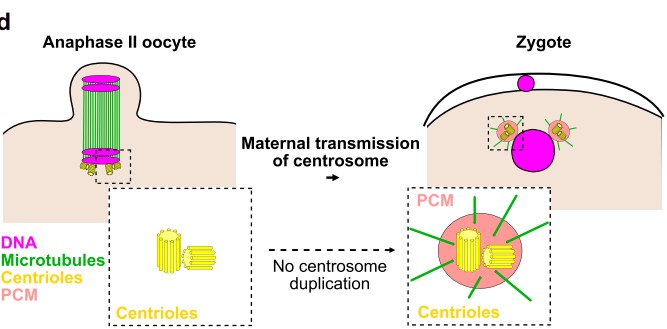

nematodes are distinctive, as maternal centrioles are not only preserved throughout the two meiotic divisions in these oocytes, but are also passed on and utilized in the zygote as building blocks for assembling functional centrosomes.

Our results raise the question of the molecular mechanism(s) behind the absence of centriole elimination in these species—whether it involves a specific protective mechanism or arises from the lack of an active elimination process. In species where centriole elimination occurs before the meiotic divisions and has been molecularly studied,

such as *Drosophila*[4], and *C. elegans*[3], a consistent sequence of events is observed, beginning with an initial loss of PCM components followed by centriole disappearance. However, the underlying mechanism and consequences of PCM removal appears to differ among species. In *Drosophila* oocytes, the PCM component Polo kinase is one of the first PCM components to depart from the oocyte centrosomes[4]. PCM components are in turn essential to maintain centriole integrity and Polo thus serves as an upstream factor that regulates the timely elimination of centrioles. Conversely, in *C. elegans* and starfish, the

**Fig. 5 | Maternal inheritance of centrioles in the parthenogenetic nematode *Diploscapter pachys*. a** Schematic (top) and immunofluorescence image (middle) of a *D. pachys* gonad. Magenta, DNA; green, *Cel*ZYG-1. Scale bar, 50 µm. Number of gonads examined is indicated on the left. Insets (bottom) show higher magnification views of single nuclei at the indicated stages. Black arrowheads indicate ZYG-1 foci. Scale bar, 5 µm. **b** Immunofluorescence images of *D. pachys* oocytes and embryos at the indicated stages. Scale bar, 10 µm. Oocyte, embryo and polar body contours are highlighted with a dashed line. Number of oocytes or embryos examined is indicated on the left for each stage. Right insets are higher magnification views of the spindle region. Magenta, DNA (all panels); green, microtubules (left and center panels); greyscale, *Cel*ZYG-1 (right panels). Scale bar, 5 µm. **c** Immunofluorescence images of *D. pachys* oocytes and embryos at the indicated stages. Scale bar, 10 µm. Oocyte and embryo contours are highlighted with a dashed line. Number of oocytes examined is indicated on the left for each stage. Right insets are higher magnification views of the spindle region. Scale bars, 5 µm and 1 µm respectively. Magenta, DNA (all panels); green, microtubule (left and center left panels); greyscale, *Cel*SPD-2 (right panels and insets). White arrowheads on insets show adjacent/engaged double foci of SPD-2 signal. **d** Schematic of the maternal transmission of two centrosomes from the oocyte to the embryo in *D. pachys*. Magenta, DNA; green, microtubules; yellow, centrioles; light red, PCM.

depletion or chemical inhibition of the Polo-like kinase orthologs, or of PCM components, does not lead to precocious centriole elimination during oogenesis[3,14,52]. We did not directly investigate the role of the Polo-like kinase ortholog in *R. diutinus* oocytes. Nevertheless, we observed that the PCM component γ-tubulin/TBG-1 disappears from centrosomes as meiosis I progresses, leaving naked centrioles at the meiotic spindle poles (Supplementary Fig. 8b) until zygotic mitosis. Assuming TBG-1 dynamics are representative of other PCM components, this result suggests that, similar to in *C. elegans*, PCM components are not necessary to maintain intact centrioles in the *R. diutinus* germline[52]. Recent findings in *C. elegans* suggest that, following PCM component removal, centriole elimination begins with the departure of SAS-1 from centrioles[31,52,64]. SAS-1 disappearance coincides with the loss of the central tube, and is accompanied by an increase in centriole diameter. We were unable to identify the SAS-1 ortholog in *R. diutinus*, and we therefore did not assess its presence or function in maintaining centriole integrity in the germline. However, our U-Ex-STED experiments demonstrated a constant centriole diameter, suggesting preservation of their structural integrity throughout meiosis in *R. diutinus* (Fig. 4b).

The presence of naked centrioles in *R. diutinus* oocytes raises another important question about their potential function as microtubule-organizing centers (MTOCs) for meiotic spindle assembly. In dividing somatic cells and spermatocytes, centrosomes are the major MTOCs that organize the mitotic spindle poles[65,66]. In oocytes of most species, acentrosomal spindle assembly entails a variety of microtubule-assembly mechanisms that originate from chromosomes, and ultimately lead to the formation of anastral spindles[6,67–70]. Thus, the absence of astral microtubules at the poles distinguishes acentrosomal spindles from centrosomal ones. The presence of polar asters during both meiotic divisions in echinoderm and mollusk oocytes suggests that the centrosomes present at the spindle poles in these oocytes are active[71]. Our immunofluorescence observations in *R. diutinus* also suggest meiotic centrosomal activity, at least during prometaphase I when astral microtubules are present (Fig. 3b, c). It would be intriguing to investigate whether the activity of meiotic centrosomes is essential for assembling a functional spindle in *R. diutinus*, echinoderm, and mollusk oocytes. An alternative, though not mutually exclusive, hypothesis is that the activity of centrosomes and the presence of polar asters may position the meiosis I spindle within the oocyte, similar to their role in mitosis in the *C. elegans* zygote[60]. Our results support this possibility. We indeed observed prometaphase I spindles positioned away from the oocyte cortex, while anaphase I consistently occurred in close proximity to the plasma membrane, suggesting relocalization of the spindle between these two cell cycle phases (Supplementary Fig. 8b, c). The progressive disappearance of the polar asters, culminating in the complete loss at the end of meiosis, correlated with the gradual decrease in centrosomal levels of the PCM protein γ-tubulin. Thus, meiosis I spindles in *R. diutinus* become progressively anastral. While we have not established a causative link between the progressive disappearance of astral microtubules and the loss of centrosomal γ-tubulin in *R. diutinus* oocytes, the ability of centrosomes to nucleate microtubules hinges on their maturation state, which is closely associated with γ-tubulin accumulation[72].

This correlation suggests a functional connection between astral microtubules and centrosomal γ-tubulin levels. Also supporting this notion, anastral meiosis II spindles formed in the same oocytes, which lack centrosomal γ-tubulin. To our knowledge, this represents the first instance of a transition from an astral to an anastral spindle within the same cell. This transition is likely important for minimizing the volume of the polar body and thus for maintaining essential maternal components in the future embryo[73].

Our findings also highlight the unusual spindle shape observed in *D. pachys* oocytes, which coincides with atypical centrosomal positioning. Typically, spindles in most cells, including oocytes, exhibit a bipolar structure. However, in the meiotic spindles of *D. pachys* oocytes, clear spindle poles could not be discerned. Instead, these meiotic spindles exhibited an apolar morphology, accompanied by the unusual positioning of centrosomes within them[54]. In contrast to *R. diutinus*, where centrosomes localized at the spindle poles, in *D. pachys* oocytes during meiosis I, centrosomes were either observed between the two homologous chromosomes or positioned on one side of the apolar spindle. As in *R. diutinus*, the role of centrosomes in nucleating meiotic spindle microtubules in the *D. pachys* oocyte is unclear. The centrosomes could behave as passive cargoes within the meiotic spindle to facilitate their segregation into the oocyte and zygote. Consistent with this hypothesis, during meiosis II both centrosomes were always observed positioned near the chromosome that was destined to be inherited by the future zygote. This striking feature underscores a key difference between the oocytes of *R. diutinus* and *D. pachys*, with implications for centrosome duplication in the zygotes of the two species. Indeed, the *R. diutinus* zygote inherits a single centrosome, necessitating its duplication before mitosis to form the two opposite spindle poles. Conversely, in *D. pachys*, the zygote receives two centrosomes, which should not duplicate in order to prevent the formation of a multipolar spindle. We suspect that the opposite fate of centrosomes between the two species stems from the different degree of engagement of their constitutive centrioles. While mother and daughter centrioles were visibly physically separated in *R. diutinus* telophase II oocytes, they remained closely apposed in *D. pachys*. As disengagement is necessary to license the initiation of the centriole duplication cycle[39], this defining feature is sufficient to explain the divergent behavior of centrosomes between the zygotes of the two species. In *C. elegans* zygotes, a physical link, possibly composed of meiotic cohesins[74], keeps mother and daughter centrioles closely connected. Separase-mediated removal of this physical link facilitates centriole disengagement and separation during the transition from meiosis to mitosis[75,76]. Investigating whether differential separase-mediated cleavage of cohesins could account for the different behavior of centrosomes between *R. diutinus* and *D. pachys* will be an interesting avenue for future research, but will require the development of tools to disrupt protein function in these species.

In *C. elegans*, embryonic polarity is established shortly after fertilization through the activity of the kinase Aurora A/AIR-1, emanating from the sperm-derived centrioles, which defines the posterior pole of the zygote[56–59]. Our findings suggest a similar mechanism likely acts as the symmetry-breaking cue in *R. diutinus* embryos. We consistently observed that the position of the posterior pole correlates with the

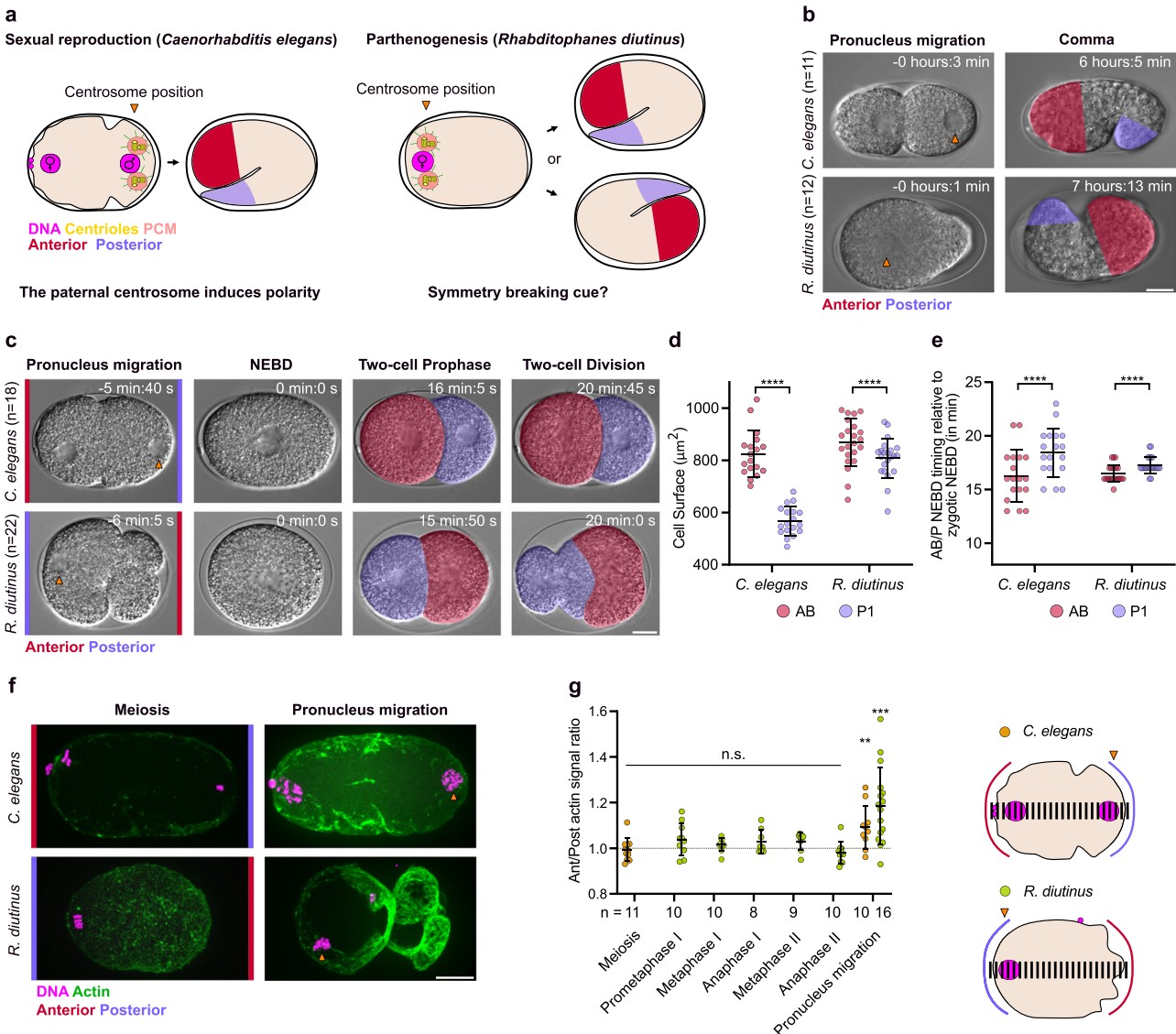

**Fig. 6 | The position of the inherited centrosome correlates with the zygote posterior pole in *R. diutinus*. a** Schematic highlighting the central role of the paternally-inherited centrosome and the potential function of its maternally-inherited counterpart in the one-cell embryo polarization of *C. elegans* (left) and *R. diutinus* (right), respectively. Magenta, DNA; yellow, centrioles; light red, centrosomes; red, embryo anterior (head); blue, embryo posterior (tail). **b** Still images from time-lapse differential interference contrast (DIC) imaging of *C. elegans* (top) and *R. diutinus* (bottom) embryos at the pronuclear migration (left) and Comma (right) stage. Anterior and posterior of Comma stage embryos are highlighted in red and purple, respectively. Orange arrowheads indicate the centrosome position. Scale bar, 10 μm. Number of embryos examined is indicated on the left for each species. **c** Still images from time-lapse DIC imaging of *C. elegans* (top) and *R. diutinus* (bottom) embryos at the indicated stages. The anterior AB and the posterior P1 blastomeres are highlighted in red and purple, respectively. Orange arrowheads indicate the centrosome position. Scale bar, 10 μm. Number of embryos examined is indicated on the left for each species. **d** Quantification of the AB (red circles) and P1 (purple circles) cell surfaces in *R. diutinus* and *C. elegans* embryos. Two-tailed paired *t*-test, alpha = 0.05, ****$p \le 0.0001$. Mean and standard deviation are displayed. **e** Quantification of the timing of nuclear envelope breakdown (NEBD) in AB or P1 relative to NEBD in the one-cell embryo of *R. diutinus* and *C. elegans*. Two-tailed paired Wilcoxon test, alpha = 0.05, ****$p \le 0.0001$. Mean and standard deviation are displayed. **f** Images of F-actin staining in *C. elegans* (top) and *R. diutinus* (bottom) oocyte and embryo at the indicated stages. Orange arrowheads indicate the centrosome position. Polar bodies are surrounded by a white line. Magenta, DNA; green, F-actin. Scale bar, 10 μm. **g** Quantification of the anterior/posterior F-actin signal ratio at the indicated stages in *R. diutinus* (green circles) and *C. elegans* (orange circles) oocytes and embryos. Number of oocytes or embryos examined is indicated below the plot for each stage. One sample two-tailed Wilcoxon test, alpha = 0.05, **$p = 0.0098$. Mean and standard deviation are displayed. Source data are provided as a Source Data file.

presence of maternal centrioles. Although because Aurora A/AIR-1 is present at the centrosomes throughout the meiotic divisions, the mechanism by which polarity is not established earlier in these oocytes remains unclear. One plausible explanation is that, similar to in *C. elegans*, the cytoplasmic pool of Aurora A/AIR-1 prevents premature polarization of *R. diutinus* oocytes[56]. In contrast in *D. pachys* oocytes, signs of polarity are already evident during the single meiotic division[54]. Although, the position of the posterior pole in *D. pachys* also correlates with the position of the meiotic spindle, the mechanism underlying

embryonic polarization is likely different from in *C. elegans* or *R. diutinus*. Indeed, unlike in *C. elegans* and *R. diutinus* zygotes, which both exhibit F-actin enrichment on the anterior cortex, *D. pachys* oocytes instead exhibit F-actin enrichment shifted toward the posterior cortex[54]. Understanding the principles and diversity of embryonic polarization mechanisms during asymmetric zygotic division in parthenogenic species is an exciting endeavor for future studies.

Overall, our results shed light on atypical and fascinating adaptations of the canonical cell division mechanisms without deleterious

effects on fitness. Studying phylogenetically distinct parthenogenetic animals presents a unique opportunity to address fundamental problems in cell biology. We suspect that the retention of maternal centrioles, as highlighted here, is inherently linked to parthenogenesis and the lack of sperm-supplied centrioles. Confirming this hypothesis would benefit from analyzing a broader range of parthenogenetic species and comparing closely related nematode species with distinct reproductive strategies. Identifying genetically related species with gonochoristic (separate sexes in different individuals), hermaphroditic, or parthenogenetic reproduction would allow for correlating the centriole transmission strategies used in these different species with specific genetic features and molecular signatures. However, understanding the molecular mechanisms that underlie these various reproductive strategies and their associated cell division adaptations will require the development of genetic tools, methods for transgenesis, and loss-of-function strategies in these underutilized model systems.

## Methods

### R. diutinus, D. pachys and C. elegans strain maintenance

The *C. elegans* N2 Bristol strain was maintained at 16 °C, 20 °C, or 23 °C. The *R. diutinus* (*sp KR. 3021*) strain (gift of Marie-Anne Félix) was maintained at 16 °C. The *D. pachys* PF1309 strain (gift of Kristin Gunsalus) was maintained at 20 °C. All nematode strains were grown under standard laboratory conditions on Nematode Growth Medium (NGM) plates (51 mM NaCl, 2.5 g Bacto Peptone, 17 g Bacto Agar, 12 μM cholesterol, 1 mM $CaCl_2$, 1 mM $MgSO_4$, 25 μM $KH_2PO_4$ and 5 μM Nystatin in 1 L dd$H_2O$ final) and fed with OP50 *E. coli* bacteria.

### R. diutinus gDNA preparation for Next Generation Sequencing (NGS)

10 young adult worms per 85 mm diameter NGM plate were incubated at 16 °C. After 6 days, worms were harvested and washed 3 times in ice-cold DNAse/RNAse-free water (Invitrogen, 10977-035). For each wash step, worms were pelleted at 4 °C by centrifugation for 3 min at 2000 g. After washes, worm pellets were partially ground using a plastic 100 μL dounce potter. For DNA extraction, the Puregen Tissue Kit was used (Qiagen, 158667). 100–200 mg of material was incubated in 5 mL of Cell Lysis Solution (Qiagen, 158906) and 25 μL of 20 mg/mL proteinase K (Qiagen, 158918) for 3 h at 55 °C, with gentle shaking every 30 min. After addition of 25 μL of 4 mg/mL RNase A (Qiagen, 158922), samples were incubated 40 min at 37 °C, then cooled down for 15 min at room temperature. Samples were incubated 5 min on ice with 1.67 mL of Protein Precipitation Solution (Qiagen, 158910). After 15 min of centrifugation at 2000 g (4 °C), the supernatant was precipitated with 5 mL of isopropanol, with 50 times slow manual tube inversion and centrifuged 5 min at 2000 g. The supernatant was removed and the gDNA pellets air-dried for 3 min, before being washed twice with 70% ethanol. Pellets were air-dried for 15 min at room temperature, then dissolved in 270–290 μL of DNA hydration solution (Qiagen, 158914). Samples were incubated 1 h at 65 °C, then at room temperature overnight, before being stored at 4 °C. For PacBio sequencing, DNA concentration was assessed with a Denovix spectrophotometer DS-11 and a QuBit dsDNA BR Assay (Life technologies, Q32850). gDNA quality was evaluated by running a 0.8% agarose ethidium bromide gel in 1X TAE. For Nanopore and Illumina sequencing, DNA concentration was assessed with a Denovix spectrophotometer DS-11 and a QuBit dsDNA HS Assay (Life technologies, Q32851). gDNA quality was evaluated by running a 1% agarose ethidium bromide gel in 1X TAE and by Agilent HS DNA analysis (5067-4626), on an Agilent 2100 bioanalyzer.

### R. diutinus gDNA sequencing

For PacBio sequencing, additional quality control was done by capillarity electrophoresis on a Femto Pulse (Agilent Technologies, Ca),

with most DNA fragments >50 kb. gDNA was sheared using a Megaruptor 2 (Diagenode) set at 25 kb. After purification of the fragmented DNA with AMpure PB magnetic beads (PacBio, 100-265-900), 10 μg of gDNA was used for library preparation using SMRTbell Express Template Prep Kit 2.0 (PacBio, 100-938-900), following manufacturer's instructions. A nuclease treatment was performed with the SMRTbell Enzyme Cleanup kit (PacBio, 101-746-400), followed by a fragment size-selection with AMpure PacBio beads (diluted at 35%) according a 3.1X cut-off (to remove <5 kb fragments). After a final quality control by Femto Pulse and QuBit dsDNA HS Assay, average fragment size of the library was 12.8 kb. Complex preparation was done using the Binding kit V 2.0 (PacBio, 101-789-500), before being loaded on a SMRT cell 8 M with a 70 pM molarity. Sequel II run was performed with 2 h of loading time, 2 h of pre-extension time and 30 h of movie time. Raw reads CCS correction was performed via the SMRTlink (version 9.0) interface with the following parameters: 3 pass and 0.99 accuracy (20.1 Gb CCS reads).

For Nanopore sequencing, library preparation was done using the Oxford Nanopore Technologies (ONT) Ligation Sequencing Kit (SQK-LSK109) following manufacturer's instructions with $2 \times 1$ μg input. Purification during library preparation was done with custom mix beads (to remove <4 kb fragments). The library quality was tested using the Qubit DNA HS Assay and 400 ng was loaded in a MinION Flow Cell (R9.4.1). Sequencing was done using the GridION device with MinKNOW 3.6.0 and base-called with Guppy 3.2.8. The run had a mean quality of 11.1 and generated 1.8 million reads with a mean length of 7.3 kb (13.6 Gb total).

For short read Illumina sequencing, library preparation was done using the Westburg NGS library Prep Kit (WB 9024) following manufacturer's instructions. Libraries were quality tested using Qubit dsDNA HS Assay and by Agilent HS DNA analysis, on an Agilent 2100 bioanalyzer. Sequencing was done on the NextSeq 550 sequencing system (Illumina, SY-415-1002) generating 78 bp paired end reads using the NextSeq 550 High Output Kit v2.5 (150 cycles) (Illumina, 20024906).

### R. diutinus genome assembly

Nanopore reads were filtered into two datasets using Filtlong v0.2.0 (https://github.com/rrwick/Filtlong), one with a target of 2.5 Gb and the second one with a target of 3.0 Gb. Both datasets were assembled using NextDenovo v2.5[77] with parameters read_type=ont input_type=raw and the outputs were combined to obtain 5 chromosome-level contigs, after discarding one contig identified as *Escherichia coli* by BLAST[78]. The contigs were polished with the PacBio HiFi reads using HyPo v1.0.3[79] with parameters -k ccs -s 50 m. Our chromosome-scale *R. diutinus* genome assembly has been deposited on NCBI Genbank.

### R. diutinus genome analyses

The previously published *R. diutinus* genome[32] was downloaded from the WormBase Parasite database (https://parasite.wormbase.org/index.html)[80] (rhabditophanes_kr3021.PRJEB1297.WBPS18.genomic). Old and new genome characteristics were analyzed with Galaxy (version 23.0.1.dev0, https://usegalaxy.org), using Quast (version 5.2.0)[81]. BUSCO v5[82] analysis, run on gVolante (https://gvolante.riken.jp/)[83], was performed on both genomes using Nematoda (3131 core genes queried) and Metazoa (954 core genes queried) ortholog sets (OrthoDB v10). Dot plot genome assembly comparison was done on D-Genies (https://dgenies.toulouse.inra.fr/)[84], with our assembly as target FASTA format using the minimap2 v2.24 (many repeats) aligner[85]. QV score was computed using meryl v1.3 (27-mer) and Merqury v1.3[86]. *k*-mer comparison plot was generated using KAT v2.4.2 (27-mer by default)[87] and the module kat comp. Haploidy score was calculated using minimap2 v2.24 to map the Nanopore reads with parameter -x map-ont and HapPy v0.1[88].

### *R. diutinus* RNA preparation for Next Generation Sequencing (NGS)

10 young adults per 85 mm diameter NGM plate were incubated at 16 °C, let grown for 5 days and rinsed with sterile M9 buffer (3 g $KH_2PO_4$, 6 g $Na_2HPO_4$, 5 g NaCl, 0.25 g $MgSO_4 \cdot 7H_2O$, in 1 L $ddH_2O$). Rinsed worms were then further cultured for 3 days in liquid suspension in S-Medium with streptomycin resistant OP50-1 *E. coli* as a food source. For preparing 1 L of S-Medium, we combined 1 L of S-Basal (5.85 g NaCl, 1 g $K_2HPO_4$, 6 g $KH_2PO_4$, in 1 L $ddH_2O$), 1 mL of 5 mg/mL Cholesterol in ethanol, 10 mL of Trace Metal Solution (1.86 g disodium EDTA, 0.69 g $FeSO_4 \cdot 7 H_2O$, 0.2 g $MnCl_2 \cdot 4 H_2O$, 0.29 g $ZnSO_4 \cdot 7 H_2O$, 0.025 g $CuSO_4 \cdot 5 H_2O$ in 1 L $ddH_2O$), 10 mL of potassium citrate pH 6.0 (26.3 g citric acid monohydrate, 268.8 g tri-potassium citrate monohydrate, in 1 L $ddH_2O$), 3 mL $MgCl_2$, and 3 mL $CaCl_2$. The resulting mixed-stage worm population was cleaned with a 60% sucrose gradient, followed by 2 washes with M9 buffer and 2 washes with 1X PBS. Cleaned adult worms were flash frozen as drops in liquid nitrogen and stored as frozen beads at -80 °C. For RNA extraction, 70–100 mg of frozen nematode beads were manually crushed using a plastic 100 μL Dounce potter in a 1.5 mL tube (Eppendorf, 0030120086). To each sample, 1 mL of TRIzol (Invitrogen, 15596018) was added followed by 10 times up and down pipetting, and 5 min incubation at room temperature. 200 μL of chloroform (VWR Chemicals, 22711.290) was added, followed by 15 times up and down pipetting. The samples were incubated 3 min at room temperature, then centrifuged 15 min at 12000 g/4 °C. The upper phase was transferred to a new RNAse free tube with 500 μL of isopropanol. After 30 gentle tube inversions and incubation on ice for 10 min, the tube was centrifuged for 10 min at 12000 *g*/4 °C. The pellets were washed with 1 mL of 75% ethanol, then centrifuged for 5 min at 7500 g/4 °C. After removing the supernatant, the pellets were air-dried for 8 min, then dissolved in 45 μL of DNAse/RNAse-free water. Samples were stored at -80 °C. RNA concentration was assessed with a Denovix spectrophotometer DS-11. Total RNA quality was determined with the Agilent RNA 6000 Nano kit (5067-1511), following the manufacturer's instructions. A RIN > 9 was set as a quality threshold for further RNA processing.

### *R. diutinus* RNA sequencing

For PacBio sequencing, cDNA was prepared using the NEBNext Single Cell/Low input cDNA Synthesis & Amplification module (New England BioLabs, 6421) following manufacturer's instructions, with 12 PCR cycles. After quality control with Qubit dsDNA HS Assay and Bioanalyzer High sensitivity (Agilent, 5067-4626), 237 ng of cDNA was used for library preparation using SMRTbell Express Template Prep Kit 2.0, following manufacturer's instructions (PacBio, 100-938-900). Complex preparation was done using the Binding kit V 2.1 (PacBio, 101-820-500), before being loaded on a SMRTcell 8 M with an 80 pM molarity. Sequel II run was performed with 2 h of loading time, 2 h of pre-extension time and 24 h of movie time. Raw reads were processed using the PacBio IsoSeq 3 pipeline (pipeline locally implemented).

For Nanopore sequencing, cDNA was synthetized with the SMART-Seq v4 Ultra Low Input RNA Kit (Takara, 634889) following manufacturer's instructions, with 10 ng RNA input. cDNA quality was verified by Agilent High Sensitivity DNA analysis (5067-4626), on an Agilent 2100 bioanalyzer. Library preparation was done using the Oxford Nanopore Technologies Ligation Sequencing Kit following manufacturer's instructions with 250 ng cDNA input. The library quality was tested using Qubit DNA HS Assay and 85 ng was loaded on a MinION Flow Cell (R9.4.1). Sequencing was done using the GridION device with MinKNOW 4.0.3 and base-called with Guppy 4.0.11. The run had a mean quality of 11.7 and generated 516,000 reads with a mean length of 752 bp (400 Mb total).

Reads for PacBio and Nanopore sequencing have been deposited on NCBI Genbank.

### Gene annotation

Repeats were identified using EDTA v1.9.8[89] with parameters --sensitive 1 --anno 1 --force 1 and the hardmasked assembly was converted into a softmasked assembly. Genes were predicted and annotated using Funannotate v1.8.9 (https://github.com/nextgenusfs/funannotate) with the Nanopore cDNA reads, combined with eggNOG-mapper version 2.1.9[90] and InterProScan v5.54-87.0[91].

### Identification of centrosomal proteins

*C. elegans* protein sequences of interest were downloaded from the Uniprot database (release 2023_02, https://www.uniprot.org). These sequences were used for performing tBLASTn (scoring matrix BLOSUM62) searches against our new chromosome-scale *R. diutinus* genome assembly using myGenomeBrowser (https://bbric-pipelines.toulouse.inra.fr/myGenomeBrowser)[92]. The lowest e-value hits were reciprocally blasted using BLASTp against the WS288 *C. elegans* database on Wormbase (https://wormbase.org/tools/blast_blat) to confirm orthology. Validity of the predicted *R. diutinus* protein sequences were further verified with the corresponding PacBio and/or Nanopore transcript reads. *C. elegans* and *R. diutinus* orthologous protein sequences were aligned with SnapGene (version 6.2.1) using a global alignment Needleman-wunsch (version 2.4.2, with a substitution matrix BLOSUM62), and exported in FASTA format. Aligned sequences are displayed using the ESPript 3.0 server (https://espript.ibcp.fr/ESPript/ESPript/index.php).

### Polyclonal antibody production and labelling

*Rdi*SAS-4, *Rdi*AIR-1 and *Rdi*ZYG-1 antibodies were custom-produced by Proteogenix (Schiltigheim, France). For *Rdi*SAS-4 and *Rdi*AIR-1, two antigenic peptides were identified (Supplementary Fig. 4a), produced with a purity above 95% and conjugated with a KLH carrier protein by Proteogenix. A 70-day immunization procedure with two rabbits per protein of interest was performed by Proteogenix, by inoculating a mix of the two purified and conjugated peptides. Antibodies were ELISA-titrated and the whole sera of both rabbits were separately affinity-purified by Proteogenix against a mix of the two peptides. For *Rdi*ZYG-1, a similar procedure was followed by Proteogenix with a single peptide inoculated per rabbit. For production of the *Rdi*TBG-1 polyclonal antibody, a single immunogenic peptide was synthetized with a purity above 75% and conjugated with a KLH carrier protein by Biomatik (Cambridge, Ontario, Canada). Rabbit immunization was performed by Eurogentec (Seraing, Belgium) following the "Speedy 28-day" program. The *Rdi*TBG-1 serum was then affinity purified "in house" by circulating the serum overnight at room temperature on a homemade Sulfolink resin column coupled with the immunogenic peptide. An acidic elution was done using 0.1 M glycine pH 2.6. Working antibody stocks were kept at −20 °C. Long-term antibody stocks were kept at −80 °C after addition of 50% v/v Glycerol (VWR Chemicals, 24388.295). Antibodies were labelled using DyLight 550 or DyLight 650 Microscale Antibody Labeling Kits (Thermo Scientific, #84531 and #84536, respectively), following the manufacturer's instructions. The dye-to-antibody ratio was estimated with a Denovix spectrophotometer DS-11.

### Oocyte and embryo staining and polyclonal antibody specificity tests

To stain oocytes and embryos, 10-20 gravid females were allowed to crawl for 5 min in a 4-8 μL droplet of Meiosis Medium (60% Leibovitz's L-15 medium (Gibco, 11415-049), 20% Heat-inactivated Fetal Bovine Serum (Sigma Aldrich, 12106 C), 0.5 mg/mL Inulin (Sigma Aldrich, I3754), 25 mM HEPES, pH 7.5) on a 76 × 26 mm glass slide (Knittel, VA111001FKB.01). Using an eyelash tool (Ted Pella, No. 1 Superfine Eyelash #113), cleaned worms were transferred into a 4 μL Meiosis Medium droplet on a glass slide coated with subbing solution (100 mL Milli-Q $H_2O$, 0.4 g Gelatin USP (Sigma, G1890), 0.04 g Chromalum (Sigma, 243361), 100 mg Poly-L-Lysine (Sigma, P-1524)). Using a

scalpel, worms were cut open at the level of the uterus and a 12 × 12 mm #1 coverslip (Marienfeld, 0101000) was gently placed on the sample. For *C. elegans* and *R. diutinus*, slides were carefully immersed in liquid nitrogen. For *D. pachys*, the freeze-crack was performed by letting slides stand on dry ice for at least 15 min. After removal of the coverslip using a razor blade, slides were immediately immersed into −20 °C cold methanol for exactly 15 min Embryos were rehydrated twice in 1X PBS for 5 min, blocked with AbDil (1X PBS, 4% BSA, 0.1% Triton X-100) for 30 min at room temperature in a homemade humidified dark chamber. For *R. diutinus*, incubation was done overnight at 4 °C with custom made DyLight 650-labeled rabbit anti-*Rdi*SAS-4, DyLight 550-labeled rabbit anti-*Rdi*TBG-1, DyLight 550-labeled rabbit anti-*Rdi*ZYG-1 or DyLight 550-labeled rabbit anti-*Rdi*AIR-1 diluted in Abdil at final concentrations of 3.32 µg/mL, 2.75 µg/mL, 4.65 µg/mL, and 1 µg/mL, respectively. For *C. elegans*, incubation was done in a similar way with Cy5-labeled rabbit anti-*Cel*TBG-1 and Cy3-labeled rabbit anti-*Cel*SAS-6[37] antibodies at a final concentration of 1 µg/mL each in Abdil. For *D. pachys*, incubation was performed overnight at 4 °C with Cy3-labeled rabbit anti-*Cel*ZYG-1 and Cy3-labeled rabbit anti-*Cel*SPD-2[37], at a final concentration of 1 µg/mL each in Abdil. After 2 washes with Abdil, a FITC-labeled mouse DM1 anti-alpha-tubulin (Sigma-Aldrich, F2168) diluted 1/100 in Abdil was incubated 1.5 h at room temperature. Finally, following 2 washes with Abdil, DNA was stained with 2 µg/mL Hoechst 33342 for 10 min at room temperature. After 2 washes with PBS, 0.1% Triton X-100 and one with 1X PBS, samples were mounted in a droplet of mounting medium (0.5% phenylenediamine in 90% glycerol, 20 mM Tris pH 8.8) on 18×18 mm #1.5 glass coverslips (Marienfeld, 0102032). Glass coverslips were sealed with nail polish and stored at −20 °C before acquisition.

To test the specificity of custom produced *R. diutinus* antibodies, we performed immunogenic peptide-blocking assays. Lyophilized immunogenic peptides with purity above 75% or 95% (Biomatik and Eurogentec) were resuspended in Milli-Q water and stored at −20 °C. Each antibody was incubated overnight at 4 °C with a 5-fold W/W excess of the corresponding immunogenic peptide under mild agitation. An equivalent volume of Milli-Q water was used as control. Subsequent immunofluorescence experiments were performed using the blocked or control antibodies as described above.

### Worm synchronization for *R. diutinus* gonad staining
A solution of S-Medium was freshly prepared and cooled-down at 16 °C. A 60 mm diameter NGM plate was placed at 16 °C. Using an eyelash tool, 10–20 gravid females were allowed to crawl for 5 min in a 20 µL droplet of S-Medium on a 76 × 26 mm glass slide, before being cut open at the uterus level with a scalpel. After dissection, one or two-cell embryos were gathered using the eyelash tool. 25 embryos were pipetted on each pre-cooled NGM plate and allowed to develop until the adult stage for 6 days at 16 °C.

### Centrosomal staining of gonads
For *C. elegans*, *D. pachys* and *R. diutinus*, the gonad staining procedure was the same as described in "oocyte and embryo staining and polyclonal antibody specificity tests", above, except that worms were cut either at the level of the tail or the head, instead of cutting worms at the level of the uterus. Primary antibody incubation was performed overnight at 4 °C. For *R. diutinus*, primary antibodies used were DyLight 650-labeled rabbit anti-*Rdi*SAS-4, DyLight 550-labeled rabbit anti-*Rdi*TBG-1 and DyLight 550-labeled rabbit anti-*Rdi*ZYG-1 all diluted 1/100 in Abdil, at final concentrations of 3.32 µg/mL, 2.75 µg/mL and 4.65 µg/mL, respectively. For *C. elegans*, the primary antibody used was a mouse anti-*Cel*IFA-1[93](a kind gift of Tamara Mikeladze-Dvali) diluted 1/50 in Abdil. For *D. pachys*, the primary antibody used was Cy3-labeled rabbit anti-*Cel*ZYG-1 at a final concentration of 1 µg/mL in Abdil. FITC-labeled mouse DM1 anti-alpha-tubulin (Sigma-Aldrich, F2168) diluted 1/100 in Abdil was incubated 1.5 h at room temperature.

For *Cel*IFA-1 staining, Alexa Fluor 488-conjugated AffiniPure Donkey Anti-Mouse (Jackson ImmunoResearch, 715-545-151) diluted 1/200 was added to the mix. Wash steps, DNA counterstaining and montage were performed as indicated in the "Oocyte and embryo staining and polyclonal antibody specificity tests" section.

### *R. diutinus* Phospho-Histone H3 (Ser10) gonad staining
10–20 females (6 days old) were allowed to crawl for 5 min in a 10 µL droplet of Egg salt medium (118 mM NaCl, 48 mM KCl, 2 mM CaCl₂, 2 mM MgCl₂, 5 mM 0.25 M HEPES pH 7.5 in ddH₂O) on a glass slide. Using an eyelash tool, clean worms were transferred in a 5 µL Egg salt medium droplet on a 76 × 26 mm glass slide coated with subbing solution. The head and/or tail were cut and a 5 µL droplet of PFA 6% (EM grade 16% solution, EMS, 15710) in Egg salt medium was applied. After pipetting up and down 10 times, incubation was performed for 5 min at room temperature in a homemade humidified dark chamber. 6 µL of solution was then carefully removed by pipetting, and a 12 × 12 mm #1 glass coverslip was placed onto the remaining drop of medium containing the fixed gonads. The montage was then immersed in liquid nitrogen. After removal of the glass coverslip with a razor blade, the glass slides were immediately immersed into 96% ethanol at room temperature for exactly 1 min. Gonads were rehydrated twice in 1X PBS for 5 min, blocked in AbDil for 30 min at room temperature. Antibody incubation was done overnight at 4 °C with the 6G3 mouse monoclonal anti-Phospho-Histone H3 (Ser10) antibody (Cell signaling, 9706 S) diluted 1/100 in Abdil. After 2 washes with Abdil, DyLight 549-conjugated AffiniPure Donkey Anti-Mouse (Jackson ImmunoResearch, 715-505-151) diluted 1/200 in Abdil was incubated 1.5 h at room temperature. Wash steps, DNA counterstaining and montage were performed as indicated in the "Oocyte and embryo staining and polyclonal antibody specificity tests" section.

### *R. diutinus* gonad spreading and staining for Phospho-MPM2
The method was adapted from ref. 31. 10–20 females (6 days old) were allowed to crawl for 5 min in a 5 µL droplet of Dissection solution (0.2X PBS, 0.1% Tween 20 Sigma, P2287). Using an eyelash tool, worms were transferred to a 5 µL droplet of Dissection solution on a 22 × 22 mm #1.5 coverslip (Marienfeld, 0107052), pre-washed in 96% ethanol for 5 min. Using a scalpel, the head and/or tail were cut and a droplet of 30 µL of Spreading solution (For 30 µL: 19.2 µL Fixative solution (4% V/V PFA (EM grade 16% solution, EMS, 15710), 3.2% sucrose W/V in Milli-Q water), 9.6 µL Lipsol ((Dutscher, 090844), 1% V/V in Milli-Q water) and 1.2 µL Sarcosyl ((Sigma Aldrich, L9150), 1% W/V in Milli-Q water) was added. Gonads were distributed over the center of a glass coverslip using an eyelash tool. The glass coverslips were allowed to dry at room temperature for 1.5 h, followed by an incubation at 37 °C for 1 h. The glass coverslips were then immersed into cold methanol at −20 °C for exactly 15 min. Gonads were rehydrated twice in 1X PBS for 5 min, blocked in AbDil for 30 min at room temperature in a humidified homemade dark chamber. Incubation was done overnight at 4 °C with a Dylight 650-labeled rabbit anti-*Rdi*SAS-4 and a mouse monoclonal anti-phospho-Ser/Thr-Pro MPM-2 antibody (Millipore, 05-368) at a final concentration of 3.32 µg/mL and 4 µg/mL respectively in Abdil. After 2 washes with Abdil, Alexa Fluor 488-conjugated AffiniPure Donkey Anti-Mouse was incubated 1.5 h at room temperature diluted 1/200 in Abdil. Wash steps, DNA counterstaining and montage were performed as indicated in the "Oocyte and embryo staining and polyclonal antibody specificity tests" section.

### EdU staining of *R. diutinus* gonads
Worms were grown on NGM plates to high density but before reaching starvation. Plates were washed with 2 mL of 1X PBS, 0.1% Triton X-100 to collect worms in a 2 mL tube. The supernatant was removed, and 500 µL of 1 mM EdU in M9 (Invitrogen Click-it EdU Alexa Fluor 488 Imaging kit, C10337) was added. The tube was covered with aluminum

and rotated vertically for 15 min at room temperature. Worms were washed once with M9 buffer, plated onto 85 mm diameter NGM plates and incubated 1 h at 16 °C. Live young adults were transferred onto two 85 mm diameter NGM plates. One plate was immediately used for the 1 h chase experiment, the other incubated at 16 °C for 28 h of chase. For both chase times, the same procedure was applied. Worms were dissected and fixed as described in "Centrosomal staining of gonads" (above). Samples were labeled with Alexa 488-azide according to the manufacturer's instructions for 1.5 h at room temperature in a humidified dark chamber. Wash steps, DNA counterstaining and montage were performed as indicated in the "Oocyte and embryo staining and polyclonal antibody specificity tests" section.

### Phalloidin staining of *R. diutinus* gonads and embryos

10–20 females (6 days old *R. diutinus* or *C. elegans*) were allowed to crawl for 5 min in a 10 µL droplet of Egg salt medium on a 76 × 26 mm glass slide. Using an eyelash tool, cleaned worms were transferred in a 5 µL Egg salt medium droplet on a glass slide coated with subbing solution. For gonad and embryo staining, worms were cut open at the level of the head and/or tail, and at the uterus, respectively. A 5 µL droplet of PFA 6% in Egg salt medium was applied to the gonad- or embryo-containing droplet. After pipetting up and down 10 times, incubation was performed for 5 min at room temperature in a humidified dark chamber. For phalloidin staining of embryos, a 12 × 12 mm glass coverslip was added, and the montage was immersed in liquid nitrogen, followed by 3 washes in 1X PBS in a coplin jar. For phalloidin staining of gonads, slides were directly washed once with 1X PBS for 5 min, followed by a 1X PBS, 0.1% Triton X-100 wash for 5 min. Samples were incubated with Phalloidin rhodamine (Thermo Fisher Scientific, R415) diluted to 4 U/mL in Abdil for 30 min at room temperature in a humidified chamber. Wash steps, DNA counterstaining and montage were performed as indicated in the "Oocyte and embryo staining and polyclonal antibody specificity tests" section.

### Phalloidin staining of oocytes

The method was adapted from ref. 94. To stain oocytes, 10–20 gravid females (*R. diutinus* or *C. elegans*) were allowed to crawl for 5 min in a 10 µL droplet of Meiosis Medium on a glass slide. Using the eyelash tool, cleaned worms were transferred into a 5 µL Meiosis Medium droplet on a glass slide coated with subbing solution. Using a scalpel, worms were cut at the level of the uterus and meiosis medium rapidly removed by pipetting. A drop of fixative (60 mM PIPES pH 6.8, 10 mM EGTA, 25 mM HEPES, 1 mM $MgCl_2$, 4% PFA (EM grade 16% solution, EMS, 15710), 0.2% glutaraldehyde (EM grade 25% solution, EMS, 16220), 0.1 mg/mL Chitinase (Sigma-Aldrich, C-6137), 100 mM D-Glucose) was added. After pipetting up and down 15 times, slides were incubated 15 min at room temperature in a humidified dark chamber. The slides were washed 3 times in 1X PBS in a coplin jar. Samples were incubated with Phalloidin rhodamine (4 U/mL in Abdil) for 1 h at room temperature. Wash steps, DNA counterstaining and montage were performed as indicated in the "Oocyte and embryo staining and polyclonal antibody specificity tests" section.

### Fixed fluorescence imaging and image processing

All fixed acquisitions were performed on a Nikon Ti-E inverted microscope, equipped with a Yokogawa CSU-X1 (Yokogawa) spinning-disk confocal head with an emission filter wheel, using a Photometrics Scientific CoolSNAP HQ2 CCD camera. The power of 100 or 150 mW lasers was measured before each experiment with an Ophir VEGA Laser and energy meter. Fine stage control was ensured by a PZ-2000 XYZ Piezo-driven motor from Applied Scientific Instrumentation (ASI). The microscope was controlled with Metamorph 7 software (Molecular Devices).

Images of gonads, oocytes and embryos were acquired with Z-sectioning every 0.1 µm, 0.2 µm or 0.3 µm using a Nikon λ APO x100/ 1.45 oil immersion objective. For gonad imaging, multiple positions were acquired to cover the surface of entire gonads. To reconstitute full gonads, gonad images were stitched using Imaris File Converter and Imaris Stitcher (Oxford Instruments) or the "Pairwise stitching" plugin in FIJI/ImageJ[95]. Displayed gonads were straightened using the FIJI/ImageJ "straighten" tool, and oriented with the distal part on the left and the proximal part on the right. For gonad, oocytes and embryos, maximum projections of relevant sections were carried out in FIJI 2.9.0/ImageJ2 1.54d. For Supplementary Fig. 6e, 3D-reconstruction was performed with Imaris 9.5.0 (Oxford Instruments). For Fig. 5a and Supplementary Fig. 7d, the DNA channel was gamma adjusted (to 0.3 and 0.1, respectively) to maximize contrast. For Supplementary Fig. 1e, 9a, c three-dimensional reconstructions were generated using the FIJI/ImageJ "3D projection" tool.

### Ultrastructure expansion STED microscopy (U-Ex-STED)

The method was adapted from ref. 31. 40–50 females were allowed to crawl for 5 min in a 5 µL droplet of Dissection solution (0.2X PBS, 0.1% Tween 20, Sigma, P2287). Using an eyelash tool, worms were transferred to a 5 µL droplet of Dissection solution on a 22 × 22 mm #1.5 coverslip (Marienfeld, 0107052), pre-washed in 96% Ethanol for 5 min. Using a scalpel, the worm head and/or tail were cut and a 30 µL droplet of Spreading solution (For 30 µL: 19.2 µL Fixative solution (4% V/V PFA (EM grade 16% solution, EMS, 15710), 3.2% sucrose W/V in Milli-Q water), 9.6 µL Lipsol ((Dutscher, 090844), 1% V/V in Milli-Q water) and 1.2 µL Sarcosyl (1% W/V in Milli-Q water)) was added. Gonads were distributed over the center of a glass coverslip using an eyelash tool. Then, the glass coverslips were incubated 15 min at room temperature in a humidified chamber. A freeze-crack was performed by incubating the coverslips on dry ice for 15 min. The glass coverslips were allowed to dry at room temperature for 1.5 h, followed by an incubation at 37 °C for 1 h. The glass coverslips were then immersed in cold methanol at −20 °C for exactly 15 min before being rehydrated twice in 1X PBS for 5 min. The glass coverslips were incubated in an Acrylamide/Formaldehyde solution (1% Acrylamide (40% Sol. Sigma, A4058) and 1% Formaldehyde (EM grade 37% solution, EMS, 15680) in 1X PBS) overnight at room temperature under mild agitation. Thereafter, glass coverslips were washed twice in 1X PBS for 5 min.

For gelation, glass coverslips were incubated in 100 µL monomer solution (19% W/W Sodium Acrylate (38% Sol. Sigma, 408220), 10% W/W Acrylamide (40% Sol. Sigma, A4058), 0.045% W/W BIS (Sigma, M1533) in 1X PBS) supplemented with 0.15% Tetramethylethylenediamine (TEMED), 0.15% Amonium Persulfate (APS) and Milli-Q water on a piece of Parafilm for 1.5 h at 37 °C in a humidified chamber in the dark. All subsequent steps were carried out at room temperature unless otherwise stated. Gels were incubated for 15 min in denaturation buffer (200 mM SDS (Sigma, L4390), 200 mM NaCl and 50 mM Tris in Milli-Q water, pH = 8.8) in 6 cm Petri dishes followed by incubation for 1 h on a 95 °C hot plate in fresh denaturation buffer. Gels were transferred to 12 cm squared Petri dishes, washed with Milli-Q water 6 times for 20 min, followed by incubation in Milli-Q water overnight at 4 °C. The expansion factor was estimated by measuring the gel size with a ruler with a 1 mm precision. After expansion, gels were blocked for 1 h in Blocking buffer (10 mM HEPES (pH = 7.2), 3% BSA (MP Biomedicals, 160069), 0.1% Tween 20, 0.04% sodium azide (Sigma, S2002), followed by incubation overnight with the primary antibody anti-RdiSAS-4 diluted in Blocking buffer at a final concentration of 1 µg/mL, under mild agitation. Gels were washed 3 times in Blocking buffer for 10 min each. Gels were incubated with the STAR RED-coupled secondary Goat Anti-Rabbit antibody (Aberrior, STRED-1002) at a final concentration of 5 µg/mL, supplemented with 2 µg/mL Hoechst 33342, diluted in Blocking buffer at 37 °C in the dark for 3.5 h and under mild agitation. Gels were washed twice in Blocking buffer for 10 min and then washed 6 times for 20 min in Milli-Q water. For imaging, gels were cut and mounted on 76 × 26 mm glass slides coated with subbing solution

(see "Oocyte and embryo staining"). A 24 × 50 mm #1.5 coverslip (Marienfeld, 0107222) was added on top of the samples and sealed with VaLaP (1:1:1 weight mix of Vaselin, Lanolin and Paraffin wax).

Acquisitions were performed on a STEDYCON microscope (2D STED microscope from Aberrior Instruments) controlled with the STEDYCON 9.0.584 software. All STED images were taken with a Zeiss 100 × 1.45NA objective, using a 640 nm excitation laser and a 775 nm depletion laser. Images of centrioles were acquired either with a 0.25 µm Z-sectioning or with a single plan. The pixel size was 20 nm, the pixel dwell time was 10 µs and the pinhole size was 64 µm. Confocal images of corresponding stages were acquired either with a 2 µm Z-sectioning or with a single plan. The two most proximal nuclei rows of the small compact nuclei were considered as the "transition zone". Displayed centriole images were deconvolved using the "DeconvolutionLab2" (https://bigwww.epfl.ch/deconvolution/deconvolutionlab2/) in FIJI/ImageJ[95] and a 0.2-pixel Median filter was applied. For Fig. 4a, 9-fold symmetrization was performed by an iterative 40° rotation of centriole images followed by a sum intensity stack projection[31].

### DIC microscopy

A solution of S-Medium was freshly prepared and cooled-down to 16 °C. The microscope room temperature was maintained at 16 °C. 10–20 gravid females (*R. diutinus* or *C. elegans*) were allowed to crawl for 5 min in a 10 µL droplet of S-Medium on a glass slide. Using the eyelash tool, cleaned worms were transferred on a glass slide in 5 µL of S-Medium and cut open with a scalpel. Embryos were transferred into 3 µL of S-Medium on a 2% agarose pad. An 18 × 18 mm #1.5 glass coverslip was gently added on the sample, sealed with softened VALAP (1:1:1 (by weight) Vaseline, lanolin, and paraffin combined with gentle heating), before being placed under the microscope. Embryos were imaged from the pronuclear migration to the 4-cell stage. Images were acquired at 5 s intervals, with 3-z planes at a step size of 1.5 µm. For long-term time-lapse imaging, dissected embryos were transferred into a 15 µL droplet of S-Medium, on a 24 × 60 mm glass coverslip (Knittel, 302980). The droplet was surrounded with a Vaseline gasket and a 18 × 18 mm #1.5 glass coverslip gently added. Embryos were imaged from pronuclear migration to hatching. Images were acquired at 20 s intervals, with 3-z planes (1-z plane for *C. elegans*) at a step size of 1.5 µm. The posterior pole is defined as the future tail in adult worms. Movies were obtained using a Nikon Eclipse Ti2 equipped with a Photometrics Prime BSI sCMOS camera using a Nikon λ APO x60/1.49 oil immersion TIRF objective, and controlled with the Metamorph 7 software (Molecular Devices).

### Image analyses and quantifications

Image analyses and quantifications were performed using ImageJ/Fiji[95]. For Supplementary Fig. 1e, homologous chromosome pairs and individual chromosomes were visually numbered. For Fig. 2f, distances were measured using the segmented line tool with a 230-pixel width on a maximum projection of the entire nuclear region. The ratio L/Ltotal was calculated by dividing the distance between the distal border of the small compact nuclei zone and the signal of interest, over the total length of the small compact nuclei zone. For Fig. 4b, the centriole diameter was determined, using the U-Ex STED images, either by measuring the external perimeter of individual centrioles imaged in end-on views with the circle tool in FIJI/ImageJ, or by measuring the longest width of individual centrioles imaged in side views with the straight-line tool in FIJI/ImageJ. Obtained values were then divided by the corresponding expansion factor. For Fig. 4c, centriole number and appearance were visually assessed and classified. For Fig. 6d, cell size was determined by measuring the two-dimensional area of each cell on single z-planes by drawing a 1-pixel width line around the contours of the AB or P cell with the "freehand selection" tool. For Fig. 6e, the timing between zygotic nuclear envelope breakdown (NEBD) and AB/P NEBD was visually determined. NEBD was assessed based on the

disappearance of the nuclear border. For Fig. 6g, average signal intensity values were obtained using a 50-pixel wide linescan drawn along the anterior-posterior axis of the medial plane of the oocytes or zygotes. The center of the linescan was used as the border between the anterior and posterior domain of the oocyte or embryo. A ratio of the average anterior intensity over the average posterior intensity was plotted. For Supplementary Fig. 9d, the distance between the segregating chromosomes was calculated on 3D projections as the length between the internal side of the two segregating chromosome sets. For Supplementary Fig. 10c, average intensity values were obtained by drawing a 50 × 50 pixel square on a 20-z stack, centered on the centrosomal *Rdi*SAS-4 signal. After cytoplasmic background subtraction, mean values were plotted.

### *R. diutinus* serial block-face-scanning electron microscopy (SBF-SEM)

10–20 young adults were allowed to crawl for 5 min in a 10 µL droplet of Milli-Q water. With an eyelash tool, cleaned worms were transferred into a 5 µL drop of Fixation solution (Glutaraldehyde 1% (EM grade 25% solution, EMS, 16220), PFA 2% (EM grade 16% solution, EMS, 15710), sodium cacodylate buffer 0.2 M pH 7.4, Milli-Q water, kept on ice during the experiment) droplet on a glass slide coated with subbing solution. Worms were cut open at the level of the tail with a scalpel, and an 80 µL droplet of Fixation solution added. Samples were incubated 30 min at room temperature in a humidified dark chamber. The head and tail of worms were further cut with a scalpel, in order to obtain a worm "cylinder" containing at least one complete gonad arm. Samples were incubated for 1 h at room temperature in a humidified dark chamber. Using an eyelash tool, worms were recovered in a 2 mL Eppendorf tube and washed 4 times with sodium cacodylate 0.2 M pH 7.4, for 5 min at room temperature. Samples were post-fixed for 2 h at 4 °C in a reduced osmium solution (2% osmium tetroxide (4% solution, EMS, 19150), 1.5% potassium ferrocyanide (Sigma Aldrich, P3289) in sodium cacodylate 0.1 M pH 7.4). Samples were washed 6 times in Milli-Q water for 3 min at room temperature followed by incubation with a 0.22 µm-filtered 1% ThioCarboxyHydrazide (Sigma Aldrich, 223220) in Milli-Q water for 40 min at room temperature. Samples were washed 6 times in Milli-Q water for 3 min at room temperature and stained with 2% osmium tetroxide in Milli-Q water for 1 h at room temperature. Samples were washed 6 times in Milli-Q water for 3 min at room temperature followed by 1% aqueous uranyl acetate incubation at 4 °C overnight. Samples were washed 6 times in Milli-Q water for 3 min at room temperature followed by a lead aspartate (0.4% L-aspartic acid (Sigma Aldrich, A9256), 0.6% Lead(II)nitrate (Sigma Aldrich, P228621)) staining 1 h at 60 °C. Samples were washed 6 times in Milli-Q water for 3 min at room temperature. Samples were then dehydrated in graded concentrations (30%, 50%, 70%, 90%, 100%) of ethanol ending in 1,2-Epoxypropane, with 20 min of incubation at room temperature for each step. Worms were infiltrated with 50% Agar low viscosity resin (Agar Scientific, R1078) in 1,2-Epoxypropane for 2 h at room temperature, 75% resin for 2 h, 100% resin for 30 min and 100% resin overnight. The resin was then changed, and samples were further incubated 2 h at room temperature, prior to inclusion by flat embedding between two 7.8 mil slides of Aclar (Aclar 33 C, 50425-10) and polymerization for 18 h at 60 °C. The polymerized blocks were mounted onto aluminum microtome stub (Oxford instruments, AGG1092450-50) for SBF-SEM imaging, with two-part conductive silver epoxy adhesive kit (EMS, 12642-14). For imaging, samples on aluminum stubs were trimmed using an ultramicrotome and inserted into a TeneoVS SEM (ThermoFisher Scientific). Acquisitions were performed with a beam energy of 2.7 kV, 400 pA or 200 pA current, in LowVac mode at 40 Pa. Spatial resolution was 12 nm with sections of 100 nm serially cut between images. Data acquired by SBF-SEM were processed using ImageJ/Fiji[95]. To compensate for jitter in the acquisition of sequential images,

image stacks from SBF-SEM acquisition were registered and rea-ligned post acquisition using the IMOD software[96].

## Figure preparation, graphs and statistical analyses

Figures and illustrations were prepared using Affinity Designer (ver. 1.10.5). Graphical representation of data and statistical analyses were performed using GraphPad Prism (version 9.4.1). Normal distribution of each dataset was checked. If at least one of the four available tests (Anderson-Darling, D'Agostino and Pearson, Shapiro-Wilk, Kolmo-gorov-Smirnov) was not positive, non-parametric tests were used. Statistical tests used are specified in the corresponding figure legends.

## Reporting summary

Further information on research design is available in the Nature Portfolio Reporting Summary linked to this article.

## Data availability

All data supporting the findings of this study are available within the paper and its Supplementary Information. Source data are provided with this paper. Reads for PacBio, Nanopore and Illumina genome and transcriptome sequencing have been deposited on NCBI GenBank under BioProject accession number PRJNA992234. Source data are provided with this paper.

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

## Acknowledgements

We thank all members of the Dumont lab for support and advice. We are grateful to Patricia Moussounda, Clarisse Picard, Téo Bitaille, and Vincent Maupu-Massamba for providing technical support. We are grateful to Marie Delattre for pointing out *Rhabditophanes diutinus* to us and for fruitful discussions. We thank Kristin Gunsalus and Marie-Anne Félix for the kind gifts of the *Diploscapter pachys* and *Rhabditophanes diutinus* strains, respectively. We thank Tamara Mikeladze-Dvali for gifting an anti-*Cel*IFA-1 antibody. We are grateful to A. Woglar and P. Gönczy for their help in setting up the U-Ex-STED protocol. Molecular graphics and analyses performed with UCSF ChimeraX, developed by the Resource for Biocomputing, Visualization, and Informatics at the University of California, San Francisco, with support from National Institutes of Health R01-GM129325 and the Office of Cyber Infrastructure and Computational Biology, National Institute of Allergy and Infectious Diseases. The ICGex NGS platform of the Institut Curie is supported by grants ANR-10-EQPX-03 (Equipex) and ANR-10-INBS-09-08 (France Génomique Consortium) from the Agence Nationale de la Recherche ("Investissements d'Avenir" program), by the ITMO-Cancer Aviesan (Plan Cancer III) and by the SiRIC-Curie program (SiRIC Grant INCa-DGOS-465 and INCa-DGOS- Inserm_12554). We acknowledge the ImagoSeine core facility of Institut Jacques Monod, member of France-BioImaging (ANR-10-INBS-04) and IBiSA, with the support of Labex "Who Am I", Inserm Plan Cancer, Region Ile-de-France and Fondation Bettencourt Schueller. This work was supported by CNRS and University Paris Cité, by NIH R01GM117407 and R01GM130764 (J.C.C.), by grant F8803-B from the Austrian Science Fund (FWF) to (A.D.), by a 4th year Ph.D. fellowship from Labex "Who Am I" ANR-11-LABX-0071 (A.P.), and by grant from the European Research Council ERC-CoG ChromoSOMe 819179 (J.D.).

## Author contributions

Conceptualization: A.P., J.D. Methodology: A.P., N.G., D.N., K.G., C.T., S.L., A.D., P.H.S., J.C.C., M.B., J.D. Investigation: A.P., J.D. Visualization: A.P., J.D. Funding acquisition: J.D. Project administration: J.D. Supervision: P.H.S., C.T., S.L., J.D. Writing – original draft: A.P., J.D. Writing – review & editing: A.P., A.D., J.C.C., J.D.

## Competing interests

The authors declare no competing interests.
