## [Peer Review File · Nature Communications]

Maternal inheritance of functional centrioles in two parthenogenetic nematodesREVIEWER COMMENTS

Reviewer #1 (Remarks to the Author):

Centrioles are a pair of cylindrical supramolecular structure found at the core of each centrosome, a major microtubule organizing center in mitotic cells. In stereotypical animal fertilization, sperm contributes centrioles, while the oocyte loses centrioles during oogenesis. However, the mode of centriole inheritance is not universal. Previous studies, including those on parthenogenetic insects capable of reproducing without sperm, have reported that oocytes can generate centriole de novo. For instance, artificially activated rat oocytes can generate centrioles without sperm involvement. Conversely, starfish and sea urchin oocytes retain centriole during oogenesis. Excess centrioles are subsequently decayed and discarded into polar bodies, whereas centrioles derived from sperm are used for later development in these organisms. In mouse oocytes, centrioles are gained from sperm, but they disintegrate, leading to acentrosomal division in early embryos. These past works have revealed that centriole inheritance is amazingly diverse.

In their manuscript, Perrier et al., document centriole inheritance in two parthenogenetic nematodes. The authors first identify several centriolar proteins in the genome of *Rabditophanes diutinus* through genome sequencing and RNAseq, and raised antibodies specific to them. This effort using non-canonical model organisms should be praised. Immunostaining with these antibodies revealed that centrioles remain intact during oogenesis and are likely inherited rather than generated de novo. Furthermore, these oocyte-derived centrioles appear to be functional and are utilized in subsequent embryogenesis, in contrast to previously known examples. Thus, this study uncovers a novel mode of centriole inheritance.

Additionally, this study highlights intriguing diversity in meiotic division mechanisms, some of which resembles those reported in other organisms such as sea urchins and starfish. These nematodes are potentially great model systems for evolutionary cell biology research into centriole inheritance if they are genome editable.

As mentioned earlier, centrioles are retained by oocytes in certain species (see a summary in the introduction of PMID 32073992 and PMIDs 11473550, 16368931). Thus, the primary discovery of this study revolves around the inheritance of "functional centriole" into embryo via oocyte, an atypical phenomenon previously unreported across organisms. In that regard, this study is an excellent addition to the field. Nonetheless, the available data largely comprise immunostaining of wild-type cells and do not definitively rule-out other possibilities. Additionally, usefulness/impact of the research and advantage of this model system over sea urchin and starfish are not clearly discussed in the manuscript. Therefore, I recommend that the authors to address the following points.

Major concerns:

1. The existence of centrioles in oocytes. SAS-4 foci may not necessarily contain actual centrioles. As seen in prior studies, the presence of centrioles in these non-canonical model systems should be confirmed through transmission electron microscopy (especially during meiosis II when SAS-4 signal is faint).
2. Uncertainty about the specificity of *R. diutinus* antibodies. To address this, the authors can consider the following options: provide ELISA data, western blotting results, or include an appropriate negative control.
3. Improvement of discussion. The manuscript's introduction primarily focused on stereotypical centriole inheritance and de novo centriole formation in parthenogenetic animals. Atypical centriole retention in sea urchins and starfish oocytes should be discussed. Additionally, the diversity in centriole segregation during meiosis of two parthenogenetic nematodes is interesting. However, similar mechanisms have been reported for other species (see above references). These parallels should be discussed. Furthermore, it appears to me that the initial paragraph in the discussion section is a part of the result section. In that case, the current discussion section contains only one paragraph. Both the limitations and implications of this research should be discussed in this section. As a non-evolutionary biologist, I would appreciate some discussions regarding strategies to utilize these models to dissect evolution.

Minor comments:

4. The title might be somewhat overstated. Future studies might unveil de novo centriole formation in other parthenogenetic nematodes.
5. Clarification of PGC. In this manuscript, the term PGC is used to describe tiny germ nuclei immediately after the gonad turn. However, it is not clear to me whether it's PGC or oogonia. In *C. elegans*, PGC refers specifically to two undifferentiated cells Z2/Z3. On the other hand, mitotic cells in the gonad are called mitotic germ cells or germline stem cells. Additionally, in Supplementary Fig5, the title is mislabeled as "progenitor germ cell."
6. Synaptonemal complex. I might be missing prior references, but I cannot find evidence supporting the recognition of the synaptonemal complex by the anti-MPM2 antibody. A relevant citation should be added.
7. Figure 5 title is an overstatement considering the fact that no functional analysis was conducted. Related to this, abstract regarding symmetry breaking cue is also an overstatement.

Reviewer #2 (Remarks to the Author):

The manuscript of Perrier et al. entitled "Centrioles are maternally-inherited in parthenogenetic nematodes" deals with an interesting and mainly unresolved aspect of the animal reproduction, the parthenogenetic development. Parthenogenesis is a kind of sexual reproduction in which the embryo develop from the female gamete without fertilization. Thus, the main problem is how the zygote inherit the centrioles need for the assembly of the mitotic spindles in the absence of the male gamete? In most cases the oocyte loses its centrosomes and the centrioles form ex novo in the egg cytoplasm after activation. In the present paper, Perrier et al. report the case of two parthenogenetic nematodes in which the centrioles are maternally inherited. This is a very interesting aspect and the AA describe this process by a lot of observations and experiments. The pictures are very nice and explicative also in consideration of the difficulty to fix and stain the very small oocytes of the nematodes. Overall, the paper is well written, the data reported are novel and results are well presented. On the basis of these considerations I retain that the paper could be published pending very few comments.

-My curiosity: the AA identify centrioles in *C.elegans* by Sas6 and in *R.diutinus* by Sas4, why the different antigens

- l.135 and Fig. 2g: Immunofluorescence staining using these antibodies revealed colocalization with SAS-4 foci throughout the *R. diutinus* gonad, from the PGCs to the -1 oocyte (Fig. 2g). I think better to add a merged image of the colocalization of SAS4 and ZYG-1 antigen, if it possible.

l. 149: These polar asters progressively disappeared as the oocyte progressed toward anaphase I, and did not reappear during meiosis II
Very intriguing!

l. 153 and Fig. 3c. SAS-4 and ZYG-1 by immunofluorescence staining. Both were present at the center of each polar aster.
Very nice pictures, but I cannot see Zyg-1 localization.

l. 169. To evaluate the nature of the centriolar structures maternally transmitted to the *R. diutinus* embryo.

I see that this section is mainly addressed to explain the recovery of diploidy.

I.208first demonstration of maternal transmission of centrioles during asexual reproduction in any species...

parthenogenesis is not asexual reproduction since the embryo develop from the female gamete, the oocyte, even if without the sperm contribution.

I. 223.in *D. pachys*, the two centrosomes remained close to each other, between or adjacent to the two chromosomes within the spindle, and did not organize the spindle poles or form astral microtubules...

How are organized the spindle microtubules in this species?

I. 235. ... *D. pachys* embryos inherit a pair of centrosomes, each comprised of two continuously engaged centrioles...

Very interesting. How can the AA explain the finding of two unseparated centrosomes each consisting of two engaged centrioles?

The Discussion sections reports additional and right questions emerging from the observation made. However I would have expected the Authors to also discuss and try to explain eventual mechanisms at the basis of the centriole inheritance in the two nematode species examined.

Reviewer #3 (Remarks to the Author):

The article by Perrier & al. examines the inheritance of centrioles in the asexual nematode species *Rh. diutinus*. The authors present compelling evidence for the maternal inheritance of centrioles, employing a meticulous and elegant cytological approach that yields stunning images. They extend this groundbreaking finding to another distantly related asexual nematode, *D. pachys*, with the same high quality of images and resolution.

Centriole number must be tightly regulated at fertilization to ensure the formation of a bipolar spindle at mitosis, for accurate chromosome segregation. Diverse strategies for centriole regulation have been documented in various sexual and asexual species. The picture so far showed that centrioles are systematically degraded during oogenesis, or soon after oocyte meiotic divisions. Consequently, in most sexual species, zygotic centrioles are paternally inherited, although a few cases involve de novo centriole formation in the zygote, post-fertilization. It has been assumed that centrioles are also formed de novo in asexual species which cannot inherit their centrioles from the sperm. However, empirical studies in asexual species are limited, leaving the question of inheritance open.

This study represents the first example, to my knowledge, of asexual species that inherit centrioles maternally. This is a breakthrough discovery which modifies our view on centriole biogenesis. With this study, we now know that the degradation of maternally provided centrioles is not a universal feature but rather a mechanism that can be turned on or off depending on the selective pressure. It also opens the intriguing possibility that maternal centriole inheritance may have been overlooked in sexual species. Furthermore, this study showcases the system's flexibility, revealing distinct inheritance strategies across two asexual nematode species.

This discovery will generate lots of excitement in the field of centriole biogenesis and evolution of asexual reproduction.

While enthusiasm is very strong for the manuscript, I believe there is room for the authors to enhance the paper further.

First, I have some reservations regarding the depth of the introduction and discussion in this study. It seems that both sections should give more background about centriole inheritance and degradation in various systems, in particular in starfish and mollusks which have not been mentioned despite their unique elimination dynamics, and perhaps exploring the broader implications of the study within the field. The discussion is a list of separate questions. It could be helpful if the authors added more details to expand on these points. As an example, two nematodes employ different strategies to maintain maternal centrioles, each with varying implications for centriole number regulation and synchronization with the S phase. This is another fascinating result of this study, but this is not discussed. See other suggestions below.

Second, the title sounds like all parthenogenic nematodes maintain their maternal centrioles. I would suggest a slight modification of the title: "Centrioles are maternally-inherited in two parthenogenetic nematode species" or "Maternal inheritance of centrioles in (two) parthenogenetic nematodes"

I also offer these suggestions:

Line 46-47: female meiosis is not systematically preceded by centrosome elimination, as shown in star-fish or clams for instance, please, modify the sentence and cite this literature

Line 60 : the end of the introduction is abrupt. Please introduce the questions and the study system

Line 137: "unlike most species where centrioles are eliminated before entering the diplotene stage", please expose this earlier in the introduction. Also, some reviews can be cited if not original articles. Again here, star fish and clams, similar to *Rh. diutinus*, maintain their centrioles passed the diplotene stage.

Line 144, 159, : "as in most species" ..see comment above, cite other studies describing alternative modes

Line 199-200: It's unclear what the authors are trying to express in this context. Two successive cell divisions with no intervening S phase, and thus no centriole duplication should lead to centrosomes with 2 centrioles in meiosis I and a single centriole at the poles in meiosis II. This is theoretically expected. Yet, this is not observed in most sperm cells for instance, which harbor 2 centrioles after the two regular meiotic divisions, or in *Rh. diutinus*.

Line 202: please define 'disengagement of centrioles' earlier in the text.

Line 202-204: this explanation will not be entirely clear for non-specialist

Line 235: please discuss this interesting finding: this is a species that inherits 4 centrioles maternally. Therefore, 8 centrioles are expected during the first mitosis.

Line 238: here or later in the discussion, please discuss the fact that these findings are in agreement with the results by Eweis & al. : the posterior pole correlates with the position of the meiotic spindle

Line 247-252: in this part of the text, or in Figure 5a, please provide more explanation, or a better figure legend. It is unclear how the posterior pole is defined. Overall, I am questioning whether Figure 5a is needed. Figure 5b seems, at least to me, sufficient to demonstrate that the position of the future posterior pole (position of P1) correlates with the position of the maternal centrioles

Line 301: please note that in *D. pachys*, homologous chromosomes are unable to pair and consequently there is no reductional meiosis. Chromosomes seem to directly enter in the second phase of anaphase, segregating chromatids away. There is no abortion of cytokinesis, or at least it has not been demonstrated by Fradin & al.

Minor corrections:

Line 54: please do not use the term "numerous" in this context since only two roles are listed (and are known so far)

Line 56 : some asexuals reproduce by sperm-dependent parthenogenesis, in which case the centrioles are inherited paternally. Maybe you could specify "strict parthenogens" here

Dear Editor and Reviewers,

We are grateful to both the Editor and Reviewers for their time, helpful comments, and rapid turn-around on the review of our paper. We appreciate this opportunity to resubmit a revised manuscript and have addressed all feasible experiments suggested by the Reviewers. As suggested by the Reviewers, we drastically edited the manuscript and the figures, and have included a point-by-point description of our response to each Reviewer comment below (blue text).

REVIEWER COMMENTS

Reviewer #1 (Remarks to the Author):

Centrioles are a pair of cylindrical supramolecular structure found at the core of each centrosome, a major microtubule organizing center in mitotic cells. In stereotypical animal fertilization, sperm contributes centrioles, while the oocyte loses centrioles during oogenesis. However, the mode of centriole inheritance is not universal. Previous studies, including those on parthenogenetic insects capable of reproducing without sperm, have reported that oocytes can generate centriole de novo. For instance, artificially activated rat oocytes can generate centrioles without sperm involvement. Conversely, starfish and sea urchin oocytes retain centriole during oogenesis. Excess centrioles are subsequently decayed and discarded into polar bodies, whereas centrioles derived from sperm are used for later development in these organisms. In mouse oocytes, centrioles are gained from sperm, but they disintegrate, leading to acentrosomal division in early embryos. These past works have revealed that centriole inheritance is amazingly diverse.

In their manuscript, Perrier et al., document centriole inheritance in two parthenogenetic nematodes. The authors first identify several centriolar proteins in the genome of *Rabbitophanes diutinus* through genome sequencing and RNAseq, and raised antibodies specific to them. This effort using non-canonical model organisms should be praised. Immunostaining with these antibodies revealed that centrioles remain intact during oogenesis and are likely inherited rather than generated de novo. Furthermore, these oocyte-derived centrioles appear to be functional and are utilized in subsequent embryogenesis, in contrast to previously known examples. Thus, this study uncovers a novel mode of centriole inheritance.

Additionally, this study highlights intriguing diversity in meiotic division mechanisms, some of which resembles those reported in other organisms such as sea urchins and starfish. These nematodes are potentially great model systems for evolutionary cell biology research into centriole inheritance if they are genome editable.

As mentioned earlier, centrioles are retained by oocytes in certain species (see a summary in the introduction of PMID 32073992 and PMIDs 11473550, 16368931). Thus, the primary discovery of this study revolves around the inheritance of “functional centriole” into embryo

via oocyte, an atypical phenomenon previously unreported across organisms. In that regard, this study is an excellent addition to the field. Nonetheless, the available data largely comprise immunostaining of wild-type cells and do not definitively rule-out other possibilities. Additionally, usefulness/impact of the research and advantage of this model system over sea urchin and starfish are not clearly discussed in the manuscript. Therefore, I recommend that the authors to address the following points.

We thanks this reviewer for their thorough assessment of our work, and for their constructive comments. We made substantial changes to the text and figures and we addressed all of this reviewer's concerns by performing every feasible experiment.

Major concerns:

1. The existence of centrioles in oocytes. SAS-4 foci may not necessarily contain actual centrioles. As seen in prior studies, the presence of centrioles in these non-canonical model systems should be confirmed through transmission electron microscopy (especially during meiosis II when SAS-4 signal is faint).

We thank this reviewer for raising this important point. Indeed, as shown recently during centriole elimination in the *C. elegans* germline¹, SAS-4 foci might correspond to centriole remnants or degenerate centrioles rather than to properly structured ones. To investigate whether intact centrioles are present throughout meiosis in *R. diutinus* oocytes, we initially attempted the approach suggested by this reviewer. However, conducting electron microscopy without a live fluorescent marker for correlating and localizing centrioles felt like searching for a needle in a haystack. Therefore, to confirm the centriolar nature of the SAS-4 foci observed throughout meiosis in *R. diutinus*, we instead employed U-Ex-STED (Ultrastructural Expansion STED microscopy)² using our custom-made *RdiSAS-4* antibody from the transition zone to telophase II oocytes. Through this method, we validated the nine-fold symmetrical structure of centrioles at every analyzed stage. Furthermore, unlike in the *C. elegans* germline where centriole diameter increases from late pachytene, marking the onset of centriole elimination with the loss of the central tube in diplotene, we did not observe a significant diameter increase in the "ring" formed by SAS-4 foci within individual centrioles. Consequently, we conclude that intact centrioles are present throughout meiosis in *R. diutinus* oocytes, and that *R. diutinus* zygotes inherit a pair of functional and structurally intact centrioles. We note that while more feasible compared to the electron microscopy approach, U-Ex-STED proved to be quite challenging in this case. The data presented here required over 30 hours of STED microscopy and analysis of 15 (approximately 10 x 10 cm) expansion gels. These data have been included in the revised version of our manuscript in Fig. 3d, e and in Supplementary Fig. 8e.

We attempted a similar experimental approach in *D. pachys*. Regrettably, the two antibodies from *C. elegans* that cross-reacted with centrosomal and centriolar components (SPD-2 and ZYG-1) did not prove compatible with the fixation procedure utilized for U-Ex-STED. Despite our numerous attempts, we were thus unable to successfully achieve the desired staining in *D. pachys*. We however note that in the *C. elegans* germline, ZYG-1 is one of the first centriolar

component that disappears almost immediately after the transition zone, and is already undetectable in pachytene³. At this stage, the centriole elimination process has already started and the structural integrity of centrioles is lost. Likewise, SPD-2 is initially visible in oocytes before cellularization, but becomes undetectable in diplotene/diakinesis oocytes. In contrast, both proteins remain visible as clear foci throughout oocyte meiosis in *D. pachys*. While we are confident that *bona fide* centrioles are maternally transmitted to *D. pachys* zygotes, we have nonetheless tempered the corresponding conclusion to accurately reflect that we have not directly demonstrated this assumption.

2. Uncertainty about the specificity of *R. diutinus* antibodies. To address this, the authors can consider the following options: provide ELISA data, western blotting results, or include an appropriate negative control.

We thank this reviewer for this important suggestion. We have now demonstrated specificity of all custom made antibodies used in this study. Specifically, we conducted immunofluorescent staining experiments in the presence or absence of the corresponding immunogenic peptides. Through this approach, we not only confirmed the specificity of each antibody but also identified the specific immunogenic peptide for each antibody. Some of our custom-made antibodies were indeed raised against two different immunogenic peptides and subsequently purified against a mix of both peptides. By conducting immunofluorescent staining in the presence of each immunogenic peptide separately, we validated that all antibodies specifically targeted a single peptide, and successfully identified each one. We have included these new data in Supplementary Fig. 4.

During the course of these experiments, we were unable to reliably replicate our initial findings regarding PKC-3 asymmetric cortical localization in the zygote (initially depicted in Supplementary Fig. 9b, c). As a result, we have opted to exclude these results from the revised version of our manuscript. It is crucial to note that this omission does not alter the overall conclusions drawn from our study.

3. Improvement of discussion. The manuscript's introduction primarily focused on stereotypical centriole inheritance and de novo centriole formation in parthenogenetic animals. Atypical centriole retention in sea urchins and starfish oocytes should be discussed. Additionally, the diversity in centriole segregation during meiosis of two parthenogenetic nematodes is interesting. However, similar mechanisms have been reported for other species (see above references). These parallels should be discussed. Furthermore, it appears to me that the initial paragraph in the discussion section is a part of the result section. In that case, the current discussion section contains only one paragraph. Both the limitations and implications of this research should be discussed in this section. As a non-evolutionary biologist, I would appreciate some discussions regarding strategies to utilize these models to dissect evolution.

We apologize for the brevity of the introduction and the discussion in the original version of our manuscript. The revised manuscript has undergone extensive editing, with both the

introduction and discussion sections completely rewritten to address all the points raised by this reviewer (and more). We hope that these revisions have greatly improved the quality and depth of our manuscript.

Minor comments:

4. The title might be somewhat overstated. Future studies might unveil de novo centriole formation in other parthenogenetic nematodes.

The title has been edited according to this reviewer's comment. The revised title is: "Maternal inheritance of functional centrioles in two parthenogenic nematodes".

5. Clarification of PGC. In this manuscript, the term PGC is used to describe tiny germ nuclei immediately after the gonad turn. However, it is not clear to me whether it's PGC or oogonia. In *C. elegans*, PGC refers specifically to two undifferentiated cells Z2/Z3. On the other hand, mitotic cells in the gonad are called mitotic germ cells or germline stem cells. Additionally, in Supplementary Fig5, the title is mislabeled as "progenitor germ cell."

We apologize for the use of this misleading terminology. We used the term PGC ('progenitor germ cells' and not 'primordial germ cells') to refer to the germline stem cells here and with regard to the fact that *R. diutinus* only produces female germ cells, but this was clearly inappropriate. We have now replaced this misleading terminology with a more accurate one based on the *C. elegans* nomenclature.

6. Synaptonemal complex. I might be missing prior references, but I cannot find evidence supporting the recognition of the synaptonemal complex by the anti-MPM2 antibody. A relevant citation should be added.

The reviewer is correct in noting that there is no published evidence supporting the recognition of the synaptonemal complex by the anti-pMPM2 antibody. In fact, this antibody targets a phospho-amino acid-containing epitope (specifically peptides containing LTPLK and FTPLQ sequences) which are present in more than 50 proteins in M-phase eukaryotic cells. Therefore, we cautiously refrained from stating that this antibody specifically recognizes the synaptonemal complex in *R. diutinus*. Instead, we mentioned that the staining pattern (five thread-like structures) obtained with this antibody resembled the synaptonemal complex. It is noteworthy that the same antibody stains a completely different structure in *C. elegans* oocytes, namely the ring-like structure found at the center of each bivalent chromosome during metaphase. Thus, the staining pattern obtained with the anti-pMPM2 antibody is context-dependent. Based on the observed pattern (five thread-like structures between DAPI-positive structures) and the karyotype of *R. diutinus* ($2n=10$), we are confident that the pMPM2 staining can serve as a readout of the pairing status of homologous chromosomes and therefore provide insights into the mitotic vs meiotic status of stained nuclei in *R. diutinus*.

7. Figure 5 title is an overstatement considering the fact that no functional analysis was conducted. Related to this, abstract regarding symmetry breaking cue is also an overstatement.

We agree with this reviewer that our previous conclusion was overstated and we apologize for this. We have revised the description of the corresponding results in both the abstract and the results section to moderate our conclusion. Additionally, we have updated the title of Fig. 5 to: "The position of the maternally-inherited centrosome correlates with the posterior pole of the zygote in *R. diutinus*."

Reviewer #2 (Remarks to the Author):

The manuscript of Perrier et al. entitled "Centrioles are maternally-inherited in parthenogenetic nematodes" deals with an interesting and mainly unresolved aspect of the animal reproduction, the parthenogenetic development. Parthenogenesis is a kind of sexual reproduction in which the embryo develop from the female gamete without fertilization. Thus, the main problem is how the zygote inherit the centrioles need for the assembly of the mitotic spindles in the absence of the male gamete? In most cases the oocyte loses its centrosomes and the centrioles form ex novo in the egg cytoplasm after activation. In the present paper, Perrier et al. report the case of two parthenogenetic nematodes in which the centrioles are maternally inherited. This is a very interesting aspects and the AA describe this process by a lot of observations and experiments. The pictures are very nice and explicative also in consideration of the difficult to fix and stain the very small oocytes of the nematodes. Overall, the paper is well written, the data reported are novel and results are well presented. On the basis of these consideration I retain that the paper could be published pending very few comments.

We thank Reviewer 2 for their appreciation of our study and we are glad they found our study interesting.

-My curiosity: the AA identify centrioles in *C.elegans* by Sas6 and in *R.diutinus* by Sas4, why the different antigens

The decision to use antibodies against SAS-6 in *C. elegans* and against SAS-4 in *R. diutinus* was based solely on their availability. Unfortunately, we were unable to identify the SAS-6 ortholog in *R. diutinus*, which prevented us from producing the corresponding antibody. Conversely, we did not have an antibody directed against the *C. elegans* SAS-4 protein. However, as both proteins are centriolar components, we are confident that the corresponding staining can be used in parallel and compared to assess the centriolar nature of the spindle poles in *C. elegans* and *R. diutinus* zygotes.

- I.135 and Fig. 2g: Immunofluorescence staining using these antibodies revealed colocalization with SAS-4 foci throughout the *R. diutinus* gonad, from the PGCs to the -1 oocyte (Fig. 2g).

I think better to add a merged image of the colocalization of SAS4 and ZYG-1 antigen, if it possible.

The corresponding merged image has been added to the revised Fig. 2.

I. 149: These polar asters progressively disappeared as the oocyte progressed toward anaphase I, and did not reappear during meiosis II

Very intriguing!

We agree with this reviewer that this observation is very intriguing. As described in our manuscript, the progressive disappearance of the polar asters correlate with the gradual decrease, culminating in complete loss, of the PCM protein γ -tubulin from the centrosomes. While we cannot definitively demonstrate a causative relationship between these observations, we strongly suspect they are linked. We now discuss this point in our revised manuscript.

I. 153 and Fig. 3c. SAS-4 and ZYG-1 by immunofluorescence staining. Both were present at the center of each polar aster.

Very nice pictures, but I cannot see Zyg-1 localization.

The sentence 'Both were present at the center of each polar aster' refers to both Fig. 3c (for SAS-4 localization) and to Supplementary Fig. 7d (for ZYG-1 localization).

I. 169. To evaluate the nature of the centriolar structures maternally transmitted to the *R. diutinus* embryo.

I see that this section is mainly addressed to explain the recovery of diploidy.

We agree with this reviewer. The sentence in question has been edited and relocated to the beginning of the next results section, which is more appropriate.

I.208first demonstration of maternal transmission of centrioles during asexual reproduction in any species...

parthenogenesis is not asexual reproduction since the embryo develop from the female gamete, the oocyte, even if without the sperm contribution.

We respectfully disagree with this reviewer. Asexual reproduction includes very diverse reproductive strategies including fission as in bacteria or some fungi, budding as in *S. cerevisiae*, vegetative propagation as in some plants, sporogenesis as in many algae, fragmentation as in planarians and many annelid worms, and agamogenesis, which includes strict parthenogenesis such as in *R. diutinus* or *D. pachys* nematodes.

I. 223.in *D. pachys*, the two centrosomes remained close to each other, between or adjacent to the two chromosomes within the spindle, and did not organize the spindle poles or form astral microtubules...

How are organized the spindle microtubules in this species?

We thank this reviewer for raising this interesting question. Unfortunately, while the behavior of centrosomes in *D. pachys* is indeed quite unusual and fascinating, we have yet to determine whether and how they participate in spindle formation in this species' oocytes. We envisage two plausible scenarios: 1) centrosomes act as passive cargoes within the meiotic spindle and are essentially "stored" within the spindle to facilitate their segregation into the oocyte and zygote, or 2) they actively nucleate microtubules that are important for spindle assembly. Despite our efforts, we have been unable to test these hypotheses due to the absence of functional tools for conducting loss-of-function experiments in this species. We have added a paragraph in the revised discussion to include this interesting point and these hypotheses.

I. 235. ... *D. pachys* embryos inherit a pair of centrosomes, each comprised of two continuously engaged centrioles...

Very interesting. How can the AA explain the finding of two unseparated centrosomes each consisting of two engaged centrioles?

We apologize for any confusion regarding this point. To clarify, our intention was not to imply that the maternally-inherited centrosomes in *D. pachys* are unseparated. Rather, each transmitted centrosome consists of two unseparated (or engaged) centrioles. We have tried to clarify this in the revised version of our discussion.

The Discussion sections reports additional and right questions emerging from the observation made. However I would have expected the Authors to also discuss and try to explain eventual mechanisms at the basis of the centriole inheritance in the two nematode species examined. We apologize for the brevity of the discussion in the initial version of our manuscript. Our revised manuscript now includes an extended discussion where we address all the points raised by this reviewer. We hope this will address the concerns raised by this reviewer.

Reviewer #3 (Remarks to the Author):

The article by Perrier & al. examines the inheritance of centrioles in the asexual nematode species *Rh. diutinus*. The authors present compelling evidence for the maternal inheritance of centrioles, employing a meticulous and elegant cytological approach that yields stunning images. They extend this groundbreaking finding to another distantly related asexual nematode, *D. pachys*, with the same high quality of images and resolution.

Centriole number must be tightly regulated at fertilization to ensure the formation of a bipolar spindle at mitosis, for accurate chromosome segregation. Diverse strategies for centriole regulation have been documented in various sexual and asexual species. The picture so far showed that centrioles are systematically degraded during oogenesis, or soon after oocyte meiotic divisions. Consequently, in most sexual species, zygotic centrioles are paternally inherited, although a few cases involve de novo centriole formation in the zygote, post-fertilization. It has been assumed that centrioles are also formed de novo in asexual

species which cannot inherit their centrioles from the sperm. However, empirical studies in asexual species are limited, leaving the question of inheritance open.

This study represents the first example, to my knowledge, of asexual species that inherit centrioles maternally. This is a breakthrough discovery which modifies our view on centriole biogenesis. With this study, we now know that the degradation of maternally provided centrioles is not a universal feature but rather a mechanism that can be turned on or off depending on the selective pressure. It also opens the intriguing possibility that maternal centriole inheritance may have been overlooked in sexual species. Furthermore, this study showcases the system's flexibility, revealing distinct inheritance strategies across two asexual nematode species.

This discovery will generate lots of excitement in the field of centriole biogenesis and evolution of asexual reproduction.

While enthusiasm is very strong for the manuscript, I believe there is room for the authors to enhance the paper further.

First, I have some reservations regarding the depth of the introduction and discussion in this study. It seems that both sections should give more background about centriole inheritance and degradation in various systems, in particular in starfish and mollusks which have not been mentioned despite their unique elimination dynamics, and perhaps exploring the broader implications of the study within the field. The discussion is a list of separate questions. It could be helpful if the authors added more details to expand on these points. As an example, two nematodes employ different strategies to maintain maternal centrioles, each with varying implications for centriole number regulation and synchronization with the S phase. This is another fascinating result of this study, but this is not discussed. See other suggestions below.

We thank Reviewer 3 for their enthusiasm and helpful suggestions and we are delighted they found our study represents a breakthrough. We agree with this reviewer that the original version of our manuscript did not introduce or discuss our findings thoroughly enough. We apologize for this and we have now extensively revised the manuscript, incorporating all the suggestions provided by this reviewer. This includes significant revisions to both the introduction and discussion sections to address this reviewer's feedback.

Second, the title sounds like all parthenogenic nematodes maintain their maternal centrioles. I would suggest a slight modification of the title: "Centrioles are maternally-inherited in two parthenogenetic nematode species" or "Maternal inheritance of centrioles in (two) parthenogenic nematodes"

We have edited the title accordingly. The revised title is: "Maternal inheritance of functional centrioles in two parthenogenic nematodes".

I also offer these suggestions:

Line 46-47: female meiosis is not systematically preceded by centrosome elimination, as shown in star-fish or clams for instance, please, modify the sentence and cite this literature
We have included all these notions and cited the corresponding literature in the revised version of the introduction.

Line 60 : the end of the introduction is abrupt. Please introduce the questions and the study system

This has been done in the last paragraph of the revised introduction.

Line 137: “unlike most species where centrioles are eliminated before entering the diplotene stage”, please expose this earlier in the introduction. Also, some reviews can be cited if not original articles. Again here, star fish and clams, similar to *Rh. diutinus*, maintain their centrioles passed the diplotene stage.

This is now introduced earlier in the introduction and the relevant literature has been cited.

Line 144, 159, : “as in most species” ..see comment above, cite other studies describing alternative modes

Done.

Line 199-200: It's unclear what the authors are trying to express in this context. Two successive cell divisions with no intervening S phase, and thus no centriole duplication should lead to centrosomes with 2 centrioles in meiosis I and a single centriole at the poles in meiosis II. This is theoretically expected. Yet, this is not observed in most sperm cells for instance, which harbor 2 centrioles after the two regular meiotic divisions, or in *Rh. diutinus*.

We apologize for this lack of clarity. Our assertion was that the sole plausible explanation for the presence of two centriolar foci at each spindle pole in Telophase II (given the context of the unsuccessful extrusion of the first polar body and thus the retention of all centriolar material in the oocyte) is that these foci were already present in meiosis I. Indeed, the centriole duplication behavior diverges dramatically between male and female meiosis. Primary spermatocytes possess a duplicated genome (4N) and four centrioles. They undergo meiosis I, resulting in the formation of two secondary spermatocytes, each containing two sets of chromosomes (2N) and two centrioles. In secondary spermatocytes, these two centrioles duplicate, independently of genome replication, to yield four centrioles before the second meiotic division, leading to a cell with four centrioles but only two sets of chromosomes (2N)^{4,5}. Hence, secondary spermatocytes demonstrate a remarkably atypical centriolar behavior, characterized by a duplication cycle that is disconnected from the S-phase. In contrast, the centriolar behavior of oocytes containing centrioles resembles that of most somatic cells, illustrated by the lack of centriole duplication observed between meiosis I and II in species such as starfish and sea urchins. The problematic paragraph has been thoroughly edited to avoid any confusion.

Line 202: please define 'disengagement of centrioles' earlier in the text.

We now define 'disengagement of centrioles' in the first paragraph of the result section in the revised version of our manuscript.

Line 202-204: this explanation will not be entirely clear for non-specialist

We have entirely rephrased this paragraph to enhance clarity.

Line 235: please discuss this interesting finding: this is a species that inherits 4 centrioles maternally. Therefore, 8 centrioles are expected during the first mitosis.

As our results strongly suggest that in *D. pachys* (and unlike in *R. diutinus*), the centrioles within each maternally-transmitted pair remain engaged, we suspect that centriole duplication is never licensed in this context. We have included these findings in the discussion section of our revised manuscript.

Line 238: here or later in the discussion, please discuss the fact that these findings are in agreement with the results by Eweis & al. : the posterior pole correlates with the position of the meiotic spindle

We have included a paragraph discussing these findings in the discussion section of our revised manuscript.

Line 247-252: in this part of the text, or in Figure 5a, please provide more explanation, or a better figure legend. It is unclear how the posterior pole is defined. Overall, I am questioning whether Figure 5a is needed. Figure 5b seems, at least to me, sufficient to demonstrate that the position of the future posterior pole (position of P1) correlates with the position of the maternal centrioles

We apologize if this was unclear. We define the posterior pole of the zygote as the one leading to the formation of the adult tail. We have tried to clarify this in the revised version of our manuscript.

Line 301: please note that in *D. pachys*, homologous chromosomes are unable to pair and consequently there is no reductional meiosis. Chromosomes seem to directly enter in the second phase of anaphase, segregating chromatids away. There is no abortion of cytokinesis, or at least it has not been demonstrated by Fradin & al.

We agree with this reviewer and we have edited this point.

Minor corrections:

Line 54: please do not use the term "numerous" in this context since only two roles are listed (and are known so far)

Corrected.

Line 56 : some asexuals reproduce by sperm-dependent parthenogenesis, in which case the centrioles are inherited paternally. Maybe you could specify “strict parthenogens” here
Done.

1. Pierron, M. *et al.* Centriole elimination during *Caenorhabditis elegans* oogenesis initiates with loss of the central tube protein SAS-1. *EMBO J* **42**, e115076 (2023).
2. Woglar, A. *et al.* Molecular architecture of the *C. elegans* centriole. *PLoS Biol* **20**, e3001784 (2022).
3. Mikeladze-Dvali, T. *et al.* Analysis of centriole elimination during *C. elegans* oogenesis. *Development* **139**, 1670-1679 (2012).
4. Rattner, J.B. Observations of centriole formation in male meiosis. *The Journal of Cell Biology* **54**, 20-29 (1972).
5. Avidor-Reiss, T. & Fishman, E.L. It takes two (centrioles) to tango. *Reproduction* **157**, R33-R51 (2019).

REVIEWERS' COMMENTS

Reviewer #1 (Remarks to the Author):

This paper represents the first demonstration of maternal inheritance of functional centrioles. The revised manuscript has been substantially improved and is a significant addition to the field. U-Ex-STED and co-staining of centrioles using antibodies against ZYG-1 and SAS-4 have provided compelling evidence supporting the maternal inheritance of centrioles in these species. Thank you for incorporating these changes, and congratulations on this discovery. I have no further comments, but below are optional considerations.

Minor comments:

#1 related to Line 278: "As we observed an identical centriolar pattern, with two disengaged centrioles visible at each disassembling spindle pole in one-cell telophase *R. diutinus* embryos (Fig. 1c), we conclude that the two maternally-inherited *R. diutinus* centrioles duplicate during the pre-mitotic S-phase, similar to the duplication observed in sperm-derived centrioles in *C. elegans*."

While I also expect centriole duplication during the pre-mitotic S-phase, the provided data does not conclusively indicate this. The authors may consider toning down the assertion in the results section.

#2 related to line 306: "Yet, unlike in *R. diutinus*, where a single pair of centrioles is maternally transmitted to the one-cell embryo, *D. pachys* embryos inherit a pair of centrosomes, each comprised of two continuously engaged centrioles (Fig. 4d)."

Although not the primary focus of the paper, the evidence here is also relatively weak. The authors may also consider toning down the assertion in the results section.

Reviewer #2 (Remarks to the Author):

My opinion is that the authors have responded comprehensively to my objections and requests. There is only one point I disagree with: the definition of parthenogenesis as asexual reproduction. However, this point has no relevance to the value of the manuscript that I believe can be published.

Reviewer #3 (Remarks to the Author):

I found that the authors adequately addressed all the concerns I had raised in my previous review. The images of expanded centrioles are stunning and provide compelling evidence to support the previous findings.

I just have two minor comments.

Line 38: 'thus , in parthenogenetic nematodes", replace by " in THESE parthenogenetic nematodes" .

Line 248: it is unclear to me why the very last sentence has been added after "retention of two sets of chromosomes and two centriolar foci"..what is the purpose of mentioning here ' producing a diploid oocyte without sperm' ?

Line 488: here is a comment that does not necessarily have to be included in the manuscript at this stage: *Bacillus* stick insects (Marescalchi, 2002) appear to form centrosomes de novo in oocytes, regardless of whether the species is sexual or asexual (in sexual species, the sperm is acentriolar). This leads to the hypothesis that such a property in sexual species has facilitated the emergence of

asexual species. Similarly, we cannot exclude that the maintenance of maternally provided centrioles is also found in some sexual species , being similarly a prerequisite for the emergence of asexuals.

We thank the editor and reviewers for their positive assessment of our revised manuscript. Here is a detailed response to their remaining concerns:

REVIEWERS' COMMENTS

Reviewer #1 (Remarks to the Author):

This paper represents the first demonstration of maternal inheritance of functional centrioles. The revised manuscript has been substantially improved and is a significant addition to the field. U-Ex-STED and co-staining of centrioles using antibodies against ZYG-1 and SAS-4 have provided compelling evidence supporting the maternal inheritance of centrioles in these species. Thank you for incorporating these changes, and congratulations on this discovery. I have no further comments, but below are optional considerations.

Minor comments:

#1 related to Line 278: "As we observed an identical centriolar pattern, with two disengaged centrioles visible at each disassembling spindle pole in one-cell telophase *R. diutinus* embryos (Fig. 1c), we conclude that the two maternally-inherited *R. diutinus* centrioles duplicate during the pre-mitotic S-phase, similar to the duplication observed in sperm-derived centrioles in *C. elegans*."

While I also expect centriole duplication during the pre-mitotic S-phase, the provided data does not conclusively indicate this. The authors may consider toning down the assertion in the results section.

We agree with this reviewer and we edited the problematic sentence: "The observation of an identical centriolar pattern, with two disengaged centrioles visible at each disassembling spindle pole in one-cell telophase *R. diutinus* embryos (Fig. 1c), suggests that the two maternally-inherited *R. diutinus* centrioles have been duplicated during the pre-mitotic S-phase, just as sperm-derived centrioles do in *C. elegans*⁵⁴."

#2 related to line 306: "Yet, unlike in *R. diutinus*, where a single pair of centrioles is maternally transmitted to the one-cell embryo, *D. pachys* embryos inherit a pair of centrosomes, each comprised of two continuously engaged centrioles (Fig. 4d)."

Although not the primary focus of the paper, the evidence here is also relatively weak. The authors may also consider toning down the assertion in the results section.

Here is the edited sentence: "Yet, unlike in *R. diutinus* where a single pair of centrioles is maternally transmitted to the one-cell embryo, our results also suggest that *D. pachys* embryos inherit a pair of centrosomes, each comprised of two continuously engaged centrioles (Fig. 5d)."

Reviewer #2 (Remarks to the Author):

My opinion is that the authors have responded comprehensively to my objections and requests. There is only one point I disagree with: the definition of parthenogenesis as asexual reproduction. However, this point has no relevance to the value of the manuscript that I believe can be published. We understand the reviewer's point; however, we believe this is primarily a semantic issue. Some consider parthenogenesis a form of asexual reproduction, while others have a stricter definition of asexuality limited to budding, fission, or fragmentation.

Reviewer #3 (Remarks to the Author):

I found that the authors adequately addressed all the concerns I had raised in my previous review. The images of expanded centrioles are stunning and provide compelling evidence to support the

previous findings.

I just have two minor comments.

Line 38: 'thus , in parthenogenetic nematodes", replace by " in THESE parthenogenetic nematodes" .
Done.

Line 248: it is unclear to me why the very last sentence has been added after "retention of two sets of chromosomes and two centriolar foci" ..what is the purpose of mentioning here ' producing a diploid oocyte without sperm' ?

We have removed the problematic part of the sentence.

Line 488: here is a comment that does not necessarily have to be included in the manuscript at this stage: Bacillus stick insects (Marescalchi, 2002) appear to form centrosomes de novo in oocytes, regardless of whether the species is sexual or asexual (in sexual species, the sperm is acentriolar). This leads to the hypothesis that such a property in sexual species has facilitated the emergence of asexual species. Similarly, we cannot exclude that the maintenance of maternally provided centrioles is also found in some sexual species , being similarly a prerequisite for the emergence of asexuals.

We thank the reviewer for these interesting observations and considerations, which will certainly inform our future thoughts on parthenogenesis.